# Adaptive laboratory evolution of microbial co-cultures for improved metabolite secretion

Dimitrios Konstantinidis[1,2] , Filipa Pereira[1,†] , Eva-Maria Geissen[1] , Kristina Grkovska[1] ,
Eleni Kafkia[1,3] , Paula Jouhten[4] , Yongkyu Kim[1,‡] , Saravanan Devendran[1] ,
Michael Zimmermann[1] & Kiran Raosaheb Patil[1,3,*]

## Abstract

Adaptive laboratory evolution has proven highly effective for obtaining microorganisms with enhanced capabilities. Yet, this method is inherently restricted to the traits that are positively linked to cell fitness, such as nutrient utilization. Here, we introduce coevolution of obligatory mutualistic communities for improving secretion of fitness-costly metabolites through natural selection. In this strategy, metabolic cross-feeding connects secretion of the target metabolite, despite its cost to the secretor, to the survival and proliferation of the entire community. We thus co-evolved wild-type lactic acid bacteria and engineered auxotrophic *Saccharomyces cerevisiae* in a synthetic growth medium leading to bacterial isolates with enhanced secretion of two B-group vitamins, viz., riboflavin and folate. The increased production was specific to the targeted vitamin, and evident also in milk, a more complex nutrient environment that naturally contains vitamins. Genomic, proteomic and metabolomic analyses of the evolved lactic acid bacteria, in combination with flux balance analysis, showed altered metabolic regulation towards increased supply of the vitamin precursors. Together, our findings demonstrate how microbial metabolism adapts to mutualistic lifestyle through enhanced metabolite exchange.

**Keywords** coevolution; experimental evolution; metabolic cooperation; multi-omics; vitamin secretion

**Subject Categories** Biotechnology & Synthetic Biology; Metabolism; Microbiology, Virology & Host Pathogen Interaction

**Mol Syst Biol. (2021) 17: e10189**

## Introduction

The long-term *Escherichia coli* evolution experiment (Lenski, 2017) has highlighted how evolution under well-controlled conditions can be used to gain fundamental insights into adaptive processes. These experiments have helped, for example, in gauging the predictability of the evolutionary outcomes (McDonald, 2019) and uncovered the divergence between the fitness trajectories and mutation rates of clonal asexual populations (Maddamsetti *et al*, 2015). Adaptive laboratory evolution is by now also a well-established tool for the development of microbial strains with improved biotechnological characteristics (Dragosits & Mattanovich, 2013). The applications include adaptation to harsh process conditions (Wallace-Salinas & Gorwa-Grauslund, 2013; Stella *et al*, 2019), improved substrate utilization (Zhou *et al*, 2012) and boosting the growth rates of metabolically engineered strains (Portnoy *et al*, 2011; Tenaillon, 2018). Further, adaptive laboratory evolution of co-cultures has enabled studying the emergence and stability of interspecies interactions such as antagonism (Koskella & Brockhurst, 2014) and metabolite exchange (Mee *et al*, 2014; Harcombe *et al*, 2018). These studies primarily focus on establishing microbial models of cross-feeding, with less emphasis on the molecular basis of adaptations. Further, while amino acid cross-feeding is common in natural communities (Machado *et al*, 2021) and has also been the basis of synthetic communities (Wintermute & Silver, 2010), cross-feeding involving other nutrients has been less well studied.

Adaptive laboratory evolution approach does not require prior knowledge of the genetic elements underlying the trait that we wish to improve. Thus, this approach can be applied to arbitrarily complex traits and to organisms that are not amenable to genetic engineering. Adaptive laboratory evolution is particularly attractive when the use of engineered organisms is restricted due to legislative or consumer preference considerations, for example in fermented food products (Burgess *et al*, 2006). The fundamental requirement for applying adaptive laboratory evolution, whether for natural or engineered organisms, is that the trait of interest is correlated to the

1   Structural and Computational Biology Unit, European Molecular Biology Laboratory, Heidelberg, Germany
2   Faculty of Biosciences, Heidelberg University, Heidelberg, Germany
3   Medical Research Council Toxicology Unit, Cambridge, UK
4   VTT Technical Research Centre of Finland Ltd, Espoo, Finland
    *Corresponding author. Tel: +44 1223 3 35640; E-mail: kp533@cam.ac.uk
    †Present address: Life Science Institute, University of Michigan, Ann Arbor, USA
    ‡Present address: Brain Research Institute, Korea Institute of Research and Technology, Seoul, South Korea

fitness under the selection conditions (Winkler *et al*, 2013). While this minimal requirement underlines the elegance and the success of adaptive laboratory evolution, it also underscores its limited applicability to the traits that impose a toll on the cell fitness, such as metabolite secretion.

To enable improvement of fitness-costly metabolite secretion while keeping the advantages offered by adaptive laboratory evolution, we here used mutualistic cross-feeding to exert selection pressure for increased production of the target compound. This approach makes the target production amenable to natural selection despite its cost to the producer. Consider a mutualistic community with two members wherein each partner depends on the other for one or more essential metabolites. Secretion of these metabolites will be directly coupled to the growth of both community members. Any or all of these compounds, despite their fitness costs for the secretors, can then be subjected to improvement via adaptive laboratory evolution by selecting for the overall community growth. We tested this concept in microbial communities consisting of lactic acid bacteria (LAB) that can naturally produce B-group vitamins (riboflavin or folate) and engineered *Saccharomyces cerevisiae* strains auxotrophic for one of the two vitamins. When grown under nitrogen excess, the yeast secretes amino acids for which the LAB strains are naturally auxotrophic (Ponomarova *et al*, 2017). The yeast-lactic acid bacterial community thus satisfies the requirement for obligate mutualism. The lactic acid bacterial strains, which are not engineered, as well as the target products, riboflavin and folate, are relevant for food biotechnological applications. Beyond this direct industrial relevance, our study establishes a proof of concept for the feasibility of improving fitness-costly traits using mutualistic communities.

## Results

### Coevolution selects for increased riboflavin secretion

Lactic acid bacteria are known vitamin secretors (Hugenholtz & Smid, 2002). We started by identifying a *Lactobacillus plantarum* strain from our natural isolate collection which could produce and secrete riboflavin. As *S. cerevisiae* is prototrophic for riboflavin, we engineered the auxotrophy by deleting *RIB4* and *RIB5* genes. This double deletion mutant yeast strain retained the wild-type ability (Ponomarova *et al*, 2017) to support the growth of *L. plantarum* in

a chemically defined medium. Thus, the community could survive in the absence of amino acids as well as riboflavin (Fig 1A).

We next performed a serial transfer adaptive laboratory evolution experiment with twelve populations derived from the same parental cultures. Non-shaking culturing conditions were used to facilitate microaerobic conditions preferred by the parental LAB strain. The non-shaking growth condition was also expected to check the emergence and/or dominance of cheater cells (i.e. cells that profit from the common goods but do not contribute in return) through facilitating spatial organization, which generally favours co-operators over cheaters (Stump *et al*, 2018). In the evolution experiment, cultures were transferred to fresh media whenever a community reached the stationary stage of growth. Following 25 transfers, the maximal optical density had increased on average sevenfold (Fig 1B). Only one of the twelve communities collapsed (after approximately 20 passages), indicating that the interactions between the partners remain generally robust. From the evolved populations, we isolated both evolved *L. planatarum* (134 isolates) and *S. cerevisiae* (6 isolates) cells for further characterization. The riboflavin secretion phenotype of 134 evolved *L. plantarum* isolates (circa 160 generations) was estimated by measuring the riboflavin fluorescence intensity in cell-free supernatant. Notably, 60% of the isolates showed improved riboflavin secretion, with up to 5.6-fold increase in the fluorescence intensity in comparison with the parental strain (Fig 1C). Four evolved bacterial isolates with increased fluorescence values, as well as one isolate with non-improved phenotype (D5), were selected for further characterization. The phenotype of these isolates was verified using ultra-performance liquid chromatography (UPLC; Appendix Table S1). Further, the amount of produced riboflavin from the isolates with improved phenotype and the parental strain, cultured as monocultures in amino acid supplemented growth medium, was quantified using liquid chromatography–mass spectrometry (LC-MS) analysis. While the parental strain secreted on average $42 \pm 16.4$ ng/ml of riboflavin, one of the tested evolved isolates, B4 (isolate B from community 4), secreted on average $134 \pm 61.1$ ng/ml (3.2-fold increase), and another one, isolate E6 (isolate E from community 6), secreted circa $426 \pm 208.7$ ng/ml (10-fold increase) (Fig 1D).

In addition to riboflavin, we also investigated whether the Flavin coenzymes, i.e. FAD and FMN, which constitute the biologically active forms of riboflavin, were secreted as well. FAD was secreted at similar levels by the parental strain and isolates B4 and E6 (Fig 1E, Appendix Table S1), but FMN was undetectable in all the

**Figure 1. Coevolution of a mutualistic community for increased production of riboflavin.**

A A natural isolate of *L. plantarum* and an engineered *S. cerevisiae* strain were cultured together for coevolution experiments. Vitamin secreting bacteria were selected based on the presence of the required biosynthetic genes in their genome. The auxotrophic *S. cerevisiae* strain was engineered by deleting RIB4 and RIB5 genes. The co-cultures were periodically transferred to new media, selecting for improved community fitness.

B Performance of the twelve co-cultures of yeast and lactic acid bacteria during the adaptive evolution experiment. The twelve communities originated from the same parental co-culture. Shown are the optical density measurements before each transfer. Individual communities are shown as blue-shaded lines; the dark blue line shows the average of the twelve communities.

C Riboflavin secretion estimates for the evolved *L. plantarum* isolates (*n* = 134) and parental strain (*n* = 5, biological replicates). Shown are the fluorescence intensity values (440-nm excitation /520-nm emission) in supernatants collected after 72 h of culturing in amino acid supplemented medium.

D Extracellular and intracellular riboflavin levels for the parental and the evolved lactic acid bacteria cultured in amino acid supplemented growth medium. The measurements are based on LC-MS analysis (*n* = 3 biological replicates).

E Levels of extracellular and intracellular flavins (FMN + FAD), for the parental and the evolved lactic acid bacteria cultured in amino acid supplemented growth medium. The measurements are based on LC-MS analysis (*n* = 3 biological replicates).

Source data are available online for this figure.

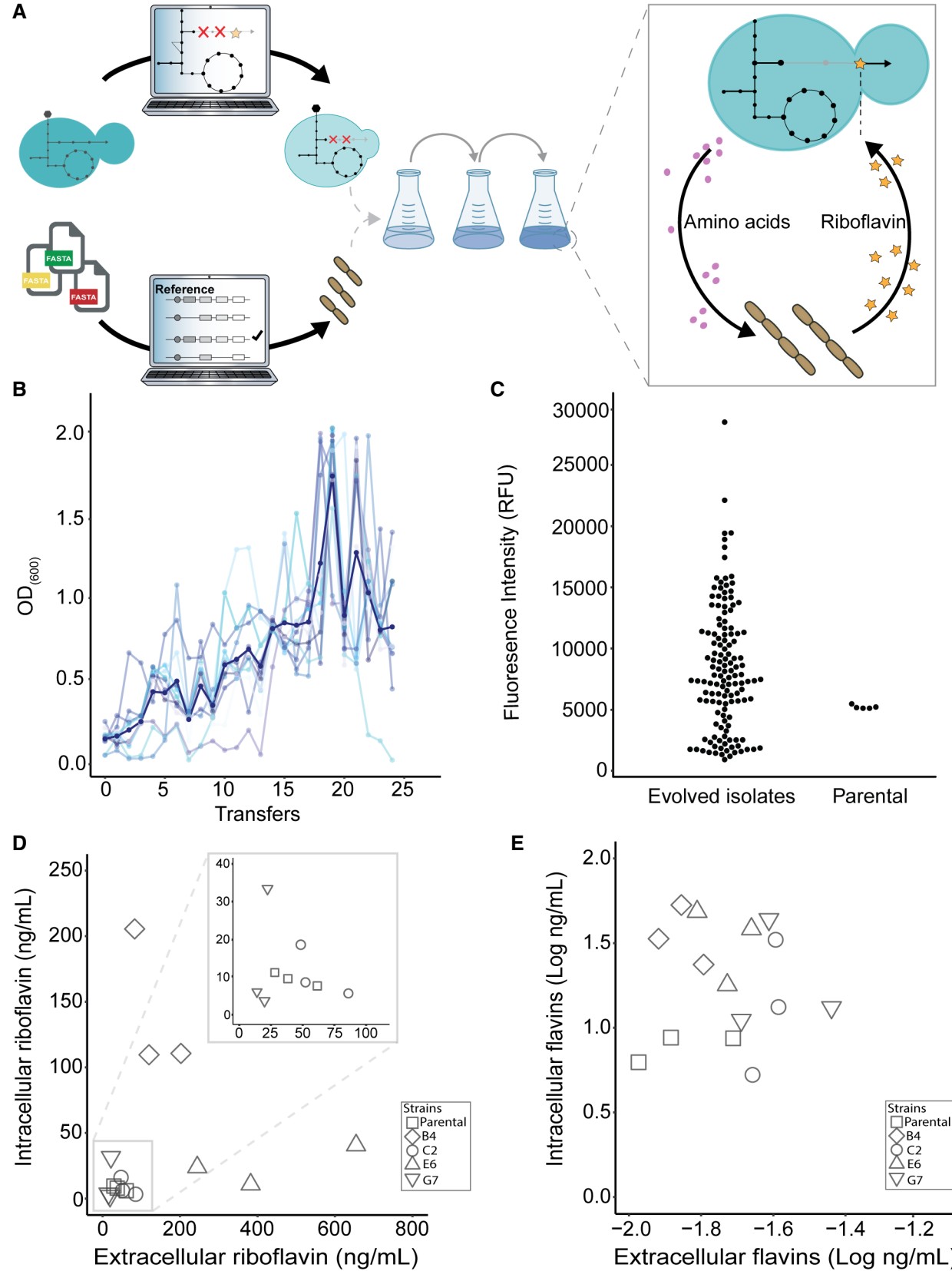

**Figure 1.**

samples. To obtain a more comprehensive picture of the biosynthetic capacity of the evolved isolates, we also analysed the intracellular accumulation of riboflavin and the flavin coenzymes. The evolved isolate B4 exhibited considerable increase in riboflavin accumulation ($144 \pm 55.1$ ng/ml; 14.4-fold increase over the parental strain; two-tailed *t*-test, $P < 0.05$) (Fig 1D). Isolate E6 accumulated $27 \pm 14.9$ ng/ml (2.7-fold increase), with a comparable increase being also observed for both FMN and FAD (Fig 1E, Appendix Table S1). In contrast to B4 and E6, the other two analysed evolved isolates (C2, isolate C from community 2, and G7, isolate G from community 7) did not show improved riboflavin secretion in comparison with the parental strain (C2: $60.9 \pm 20.7$ ng/ml, G7: $18.2 \pm 3.2$ ng/ml). However, these isolates secreted higher amounts of FAD (1.7 and 1.9-fold increase for C2 and G7, respectively; two-tailed *t*-test, $P < 0.05$) and had similar levels of intracellular riboflavin as the parental strain (C2: $11 \pm 6.7$ ng/ml, G7: $15 \pm 16.6$ ng/ml, Fig 1E). Together, growth improvement and increased riboflavin accumulation and secretion show that the coevolution resulted in positive selection for increase in vitamin biosynthesis.

### Lactic acid bacteria improved amino acid utilization

Our coevolution setting links increased production of a desired compound to the improved fitness of the whole community, even if the increase in productivity imposes a burden on the individual cell fitness. Indeed, both isolates with improved producing phenotype, B4 and E6, exhibited a strong decrease in their growth rate in comparison with the parental when grown as monocultures in media with supplemented amino acids (B4: two-tailed *t*-test, $P < 0.05$, E6: two-tailed *t*-test, $P < 0.05$; Fig 2A). When we mimicked the co-culture conditions by culturing the evolved bacterial isolates under limited nitrogen availability (Ponomarova *et al*, 2017), the parental strain's $OD_{600}$ value and growth rate significantly decreased (two-tailed *t*-test, $P < 0.05$), while all evolved isolates had similar or improved growth rate (Fig 2B). Additionally, co-cultures of the parental yeast and improved bacterial isolates grew better than the co-cultures involving either only the parental strains or the evolved "negative control" bacterial isolate D5 (isolate with decreased riboflavin production; Appendix Table S1, Fig 2C). To monitor closer the species dynamics which occurred in this mutualistic community, flow cytometry analysis was performed in co-cultures containing a Δ*rib4:rib5* yeast strain, engineered to constitutively express a red fluorescence protein (RFP), and either the parental *L. plantarum* strain or one of the evolved bacterial isolates. Sorting of the co-culture populations showed that the evolved isolates B4, C2 and E6 supported 1.5- to 3.5-fold more yeast cells than their parental strain (two-tailed *t*-test, $P < 0.05$; Fig 2D), while isolate D5 could still support a number of yeast cells similar to the parental (Fig 2D). Per co-culture OD, the number of bacteria somewhat increased, albeit with considerable variation across replicates (Fig EV1A). However, the ratio of the bacterial to yeast cells did not change significantly (two-tailed *t*-test, $P > 0.05$), in line with the mutualistic nature of the yeast-lactic acid bacterial relation whereby the growth of both species is tightly linked (Fig EV1B). By secreting higher amount of riboflavin, the evolved isolates provided more resources to their yeast partner, which, in turn, provided amino acids to the bacteria.

### Direct contact with yeast is not necessary for vitamin production by lactic acid bacteria

Lactic acid bacterial and yeast cells can co-aggregate to form interspecies biofilms (Kawarai *et al*, 2007; Arroyo-López *et al*, 2012). We therefore set out to investigate whether the improved secretion through coevolution was dependent on direct cell–cell contact and/or on the spatial structure of the community. Microscopic observation of the co-cultures between either the parental *L. plantarum* or the evolved isolate E6, and the parental Δ*rib4:rib5 S. cerevisiae* showed formation of similar extended aggregates (Fig EV1C and D). We next compared the aggregation and the biofilm forming phenotype of the bacterial isolates to determine whether improved secretion is associated with specific cell surface traits that promote cell contact. Evolved isolates C2 and E6 could aggregate more than the parental strain, both in the presence of yeast and in the monoculture, while isolate B4 was aggregating less, independently of the presence of yeast. Nevertheless, the observed differences were small and not statistically significant (Fig EV1E). Furthermore, all three isolates showed significantly higher tendency for biofilm formation in monoculture in comparison with their parental strain (two-tailed *t*-test, $P < 0.05$), but this change was not observed in the corresponding co-cultures (Fig EV1F). These experiments indicate that there was no selection pressure towards increased aggregation between bacteria and yeast.

The independence of increased riboflavin secretion from cell–cell contact was further supported by the observation that bacterial conditioned medium could sufficiently support yeast growth and vice versa. The growth rate of yeast cultures in conditioned medium of evolved bacterial isolates increased on average by 20% and the maximal optical density by 40% (Fig 2E). Similarly, the growth of the evolved bacterial isolates considerably improved compared to their parental strain when cultured in medium conditioned by the wild-type yeast (two-tailed *t*-test, $P < 0.05$; Fig 2F). Furthermore, both the parental strain and the evolved bacterial isolate E6 displayed improved fitness in the conditioned medium of the evolved yeast isolates, with 16 and 49% increase in the optical density, respectively (Fig EV2A and B). When these evolved yeast isolates were cultured in conditioned medium from the parental *L. plantarum* (Appendix Fig S1A) and the evolved bacterial isolate E6 (Appendix Fig S1B), they exhibited increased maximal optical density compared to the parental Δ*rib4:rib5* strain, as well as increased growth rate (~34% on average). Both the growth fitness and the conditioned media experiments confirm the intended natural selection for increased vitamin secretion and the lack of selection pressure towards increased aggregation. Bacterial isolates with an over-secreting phenotype emerged in the population and were able to enhance growth of their partner by supplying the essential vitamin. At the same time, the evolved isolates exhibited increased capability to assimilate the resource offered by their partner, resulting in net positive impact on the community growth. The increased cost of riboflavin production was thus more than compensated for, through more efficient utilization of amino acids secreted by the yeast.

### Regulation of biosynthetic pathway and precursor supply underlines increased vitamin production

Following the successful coevolution, we set out to identify the molecular changes underlying the improved riboflavin secretion.

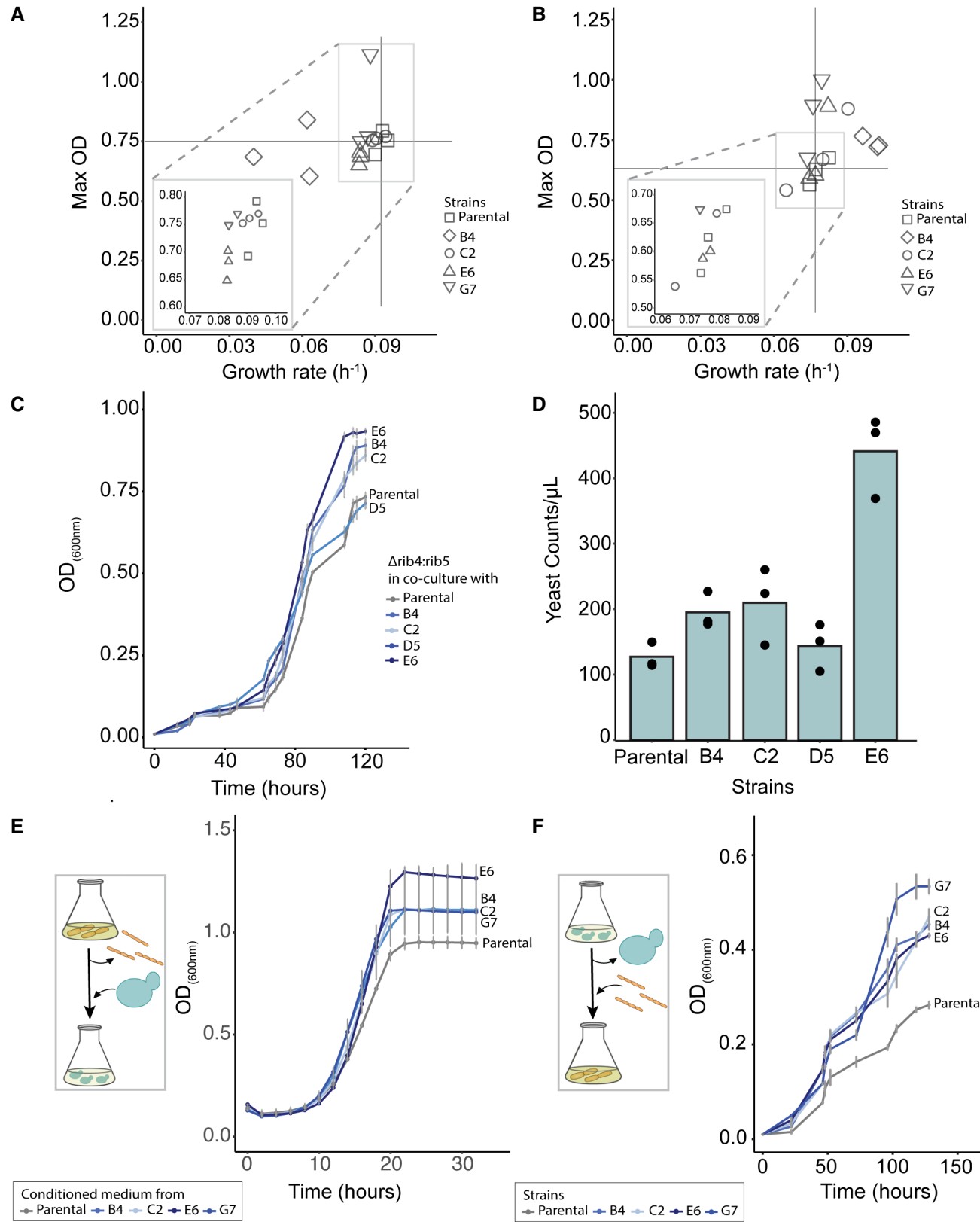

**Figure 2.**

◀

**Figure 2.  Effect of coevolution on the growth fitness.**

A   The growth rates and the maximum optical density reached by the parental and the evolved *L. plantarum* isolates cultured in amino acid supplemented medium. The vertical and the horizontal lines mark the average growth rate and optical density of the parental strain, respectively (*n* = 3 biological replicates).

B   The growth rates and the maximum optical density reached by the parental and the evolved *L. plantarum* isolates cultured under limited amino acid supplementation. The vertical and the horizontal lines mark the average growth rate and optical density of the parental strain, respectively (*n* = 3 biological replicates).

C   Growth kinetics of the parental *L. plantarum* and four evolved isolates co-cultured with *S. cerevisiae* engineered for riboflavin auxotrophy (*n* = 3 biological replicates; grey bars show mean ± SD).

D   Yeast cell estimates (RFP-positive events in fluorescence-activated cell sorting, Methods) in co-cultures as in panel C. Samples were collected at 84 h (*n* = 3 biological replicates).

E   Growth kinetics of the parental Δrib4:rib5 *S. cerevisiae* strain in conditioned medium of the parental and evolved bacterial isolates (*n* = 3 biological replicates; grey bars show mean ± SD).

F   Growth kinetics of the parental and evolved bacterial isolates in medium conditioned by wild-type *S. cerevisiae* (*n* = 3 biological replicates; grey bars show mean ± SD).

Source data are available online for this figure.

Genome sequencing of the evolved bacterial isolates did not reveal any mutations along the riboflavin operon, but we identified single nucleotide polymorphisms (SNPs) in various genes that regulate transcription or are associated with the uptake of amino acids (Appendix Table S2). Evolution studies on obligatory mutualistic microorganisms have reported similar findings (Turkarslan *et al*, 2017) indicating that the fine-tuning of regulatory elements is required for optimizing the usage of resources provided by mutualistic partners. Since none of the evolved isolates exhibited any mutations with direct link to vitamin biosynthesis, we decided to analyse the changes to protein and metabolite abundances.

Through proteomic analysis of bacterial monocultures under amino acid supplementation, we identified 55 proteins that were significantly different between the parental and the evolved bacterial isolates (FDR-adjusted $P < 0.05$, $\log_2$fold change $\leq -1$ or $\geq 1$). In accord with the genome sequencing results, these proteins included both transcriptional regulators and amino acid transporters (Fig EV3A, Appendix Table S3). Several proteins involved in the biosynthesis of purines and riboflavin were also found to be differentially abundant, providing a more direct link to the increased production. All evolved isolates exhibited higher abundance of RibA (GTP cyclohydrolase-2), which catalyses the first reaction of the riboflavin biosynthetic pathway (Abbas & Sibirny, 2011). An increase, albeit less prominent, in the abundance of other Rib proteins, such as RibD (isolates B4 and E6, 0.9 and 0.7 $\log_2$fold change, respectively) and RibB (isolates C2 and G7, 0.7 and 1.1 $\log_2$fold change, respectively) was also observed. Thus, riboflavin biosynthesis seems to be largely controlled at the first enzymatic reaction catalysed by RibA and by the supply of the precursors originating from the purine biosynthesis pathway. To put the proteomic changes in the context of the overall metabolic network, we used the genome-scale metabolic model of *L. plantarum* for estimating the metabolic fluxes in the parental strain and the evolved isolates that best fit to the relative metabolic enzyme levels (Methods). Specifically, we performed a Metabolic Analysis with Relative Gene Expression (MARGE, Methods), which utilizes linear optimization to predict optimal metabolic states that best fit the relative changes in protein abundances, while the total metabolic proteome level is maintained. Through MARGE, we predicted increased flux to the riboflavin pathway for all the evolved isolates, showing that the observed changes in the metabolic enzyme abundances are consistent with the increased vitamin production.

Gas chromatography–mass spectrometry (GC-MS) analysis of the exometabolome, originating from the same bacterial monocultures used for the proteomics analysis, showed that apart from the increased riboflavin secretion, all the evolved isolates maintained similar secretion profile as the parental strain. On the other hand, analysis of the intracellular metabolites revealed additional differences associated with riboflavin biosynthesis (Fig EV3B, Appendix Table S4). These include nucleotide metabolism as well as glycolysis and the pentose phosphate pathway. For example, we detected increased levels of ribose-5-phosphate and of glucose-6-phosphate, which lead to formation of GTP, a vital precursor of the riboflavin pathway (Abbas & Sibirny, 2011).

Interestingly, the intracellular levels of glutamine and glutamate, amino acids closely linked to riboflavin biosynthesis (Burgess *et al*, 2004), were also increased in the evolved bacteria. In contrast, the levels of amino acids that are not associated with the riboflavin pathway, such as alanine, were unaltered. The increase in intracellular amino acid levels is also in accord with the yeast-LAB co-dependencies, as glutamine, glutamate and alanine are among those secreted by the yeast (Ponomarova *et al*, 2017). Following the observed differences in intracellular amino acid levels in the evolved bacterial isolates, we asked whether the secretion phenotype of the yeast also altered during the evolution. Indeed, all the evolved yeast isolates secreted higher amounts of glutamate and aspartate (two-tailed *t*-test, $P < 0.05$; Fig EV2D and G), with two of the isolates (EY1 and EY5) secreting higher amounts of alanine as well (two-tailed *t*-test, $P \leq 0.05$; Fig EV2C). On the other hand, all isolates secreted lower amounts of glutamine (two-tailed *t*-test, $P < 0.05$; Fig EV2F), while the levels of threonine and serine remained largely unchanged (Fig EV2E and H). Together, increased availability of glutamate, upregulation of the riboflavin biosynthetic pathway and increased precursor supply from the pentose phosphate pathway likely contributed to the increased riboflavin production by the evolved bacteria.

**Coevolution for folate production**

To confirm the generality of our coevolution approach, we addressed production of an additional vitamin, viz., folate. The parental *L. plantarum* strain used in the riboflavin case also naturally produces folate, albeit in small amounts. As in the riboflavin case, we co-evolved this wild-type *L. plantarum* with a *S. cerevisiae*

strain engineered for folate auxotrophy ($\Delta abz1$). Following the adaptive laboratory evolution, the optical density increased by circa 5-fold (Appendix Fig S2A) and the amount of folate secreted, among the tested bacterial isolates, increased by 3-fold on average; the best secreting evolved isolate (B8) produced $190 \pm 35.6$ ng/ml of folate, while the parental strain secreted $48.3 \pm 10.5$ ng/ml of folate (Fig 3A). When grown in monocultures, in medium supplemented with amino acids, all the evolved isolates grew to higher maximum $OD_{600}$ than their parental strain, and isolates A3 and C9 also exhibited higher growth rates (Fig 3B). Similar to the riboflavin case, the parental folate auxotroph yeast ($\Delta abz1$) showed improved growth in the conditioned medium of the evolved bacterial isolates with improved secretion (B8 and H1), while all the evolved bacterial isolates grew better than their parental when cultured in the yeast conditioned medium (Appendix Fig S2B and C). The improved secretion in coevolution is thus not limited to a particular compound. In both cases (riboflavin and folate), improved secretion phenotypes in bacteria evolved alongside with more efficient utilization of the amino acids secreted by their yeast partner.

## Coevolutionary selection is due to cross-feeding and generalizable to other species and products

We next asked whether the observed improvements in vitamin secretion could be partly due to other factors, such as the metabolic effect of the growth conditions or regulatory predisposition of the lactic acid bacteria. To address this, we performed a control experiment whereby the parental *L. plantarum* strain was evolved in the absence of both yeast and riboflavin, but under amino acid supplementation (Appendix Fig S3A). Quantification of the riboflavin in the cell-free supernatant using the fluorescence assay showed that most evolved isolates (59%) performed worse (1.2- to 4-fold decrease) than the parental. Moreover, the average increase for the isolates evolved in monoculture was only 1.2-fold, twice lower than the change observed in coevolution (Fig 3C). These monoculture evolution experiments further confirm that the selection pressure created by metabolic cross-feeding was indeed the driver for the improved secretion emerging in the coevolution.

We next examined whether the overproducing phenotype emerging in coevolution is specific for the target compound or the selection pressure leads to unspecific changes in related pathways. For this, we estimated the riboflavin producing capabilities of isolates originating from the coevolution for increased folate production. We detected a small increase in fluorescence values ($1.05 <$ fold change $< 2$) for about half of these isolates (49%). However, this increase is significantly lower (circa 3.5-fold lower; two-tailed *t*-test, $P < 0.05$) compared to that for the isolates selected from the co-cultures evolved for increased riboflavin production (Fig 3C). We also investigated whether the coevolution can be generalized to other species beyond *L. plantarum*. We thereby co-evolved two additional species, viz., *Lactococcus lactis* and *Brevibacterium casei*, with the riboflavin auxotroph *S. cerevisiae* (Appendix Fig S3B and C). As expected, evolved isolates from both communities exhibited increased riboflavin production with up to 5.6-fold increase in the fluorescence readout (Fig 3D). Thus, as long as the conditions for the obligate mutualism are met, community evolution can be generalized to different products and species.

## Agent-based model supports fitness advantage of increased secretion

We were able to capture the key aspects of the coevolution experiments in a simulation study using an agent-based model. We were interested in understanding why the cheaters, i.e. bacterial cells that secreted low amounts or no vitamin but utilized the common goods (amino acids secreted by yeasts), did not lead to the population collapse. We simulated the fate of the mutants with an improved secreting phenotype but slower growth and studied the effect of increased mixing of the liquid culture on the mutant's chance to persist. First, we simulated a scenario of genetic drift where the mutant secretes the same amount of vitamin as the wild type, to establish a baseline to which the other simulations can be compared. Next, we repeated this simulation setup for different degrees of increase in secretion. We observed that the chance of a better secreting mutant to invade, i.e. to still be present after 25 transfers, was significantly higher (Fisher's exact test, $P < 0.05$) for low mixing strengths. This was observed for all simulated degrees of increased secretion (Fig EV4A). In the absence of mixing, the chance of successful invasion was tenfold higher for a mutant with increased secretion than expected by chance (Fisher's exact test, $P < 0.05$) (Fig EV4A). Nevertheless, with increasing mixing strength, the invasion frequency first dropped below that expected by random (Fisher's exact test, $P > 0.05$), and then to zero. The absolute values for the chance of invasion did not vary significantly between the mutants with different degrees of secretion increase (Fig EV4A). In addition to the chance of invasion, we assessed the fraction of the mutant in the total bacterial population at the end of the simulation in successful invasions. We found that this fraction depended on how strongly the secretion was increased, with higher secretion resulting in lower fractions of the mutant. The mutant fraction also depended on the strength of mixing, with stronger mixing resulting in lower mutant fractions down to total extinction of the mutant bacteria. The simulation results corroborate that a mutant with increased secretion but slower growth can have a selection advantage in an obligate mutualistic community in the absence of mixing (Fig EV4B–E).

## Evolved bacteria retain the improved secreting phenotype under application-relevant conditions

We next asked whether the enhanced riboflavin secretion phenotype of the evolved isolates would be retained under different, application relevant, growth conditions, and in a medium where riboflavin is freely available. We therefore tested the vitamin secretion in milk, which is relevant in several food applications such as yogurt, kefir and cheese production. When we inoculated either the parental *L. plantarum* or one of the evolved isolates (B4 and E6), the concentration of riboflavin increased above the levels detected in the milk at the start. While riboflavin concentration increased by 1.25-fold in cultures of the parental strain, the evolved isolates B4 and E6 showed 1.35- and 1.46-fold increase (B4: two-tailed *t*-test, $P < 0.05$, compared to milk control, $P = 0.15$ compared to the parental strain, E6: two-tailed *t*-test, $P < 0.05$ compared to milk control, $P < 0.05$ compared to the parental), respectively (Fig EV5). These results confirm the genetic imprinting of the coevolution, with the evolved isolates retaining their phenotype in the absence of

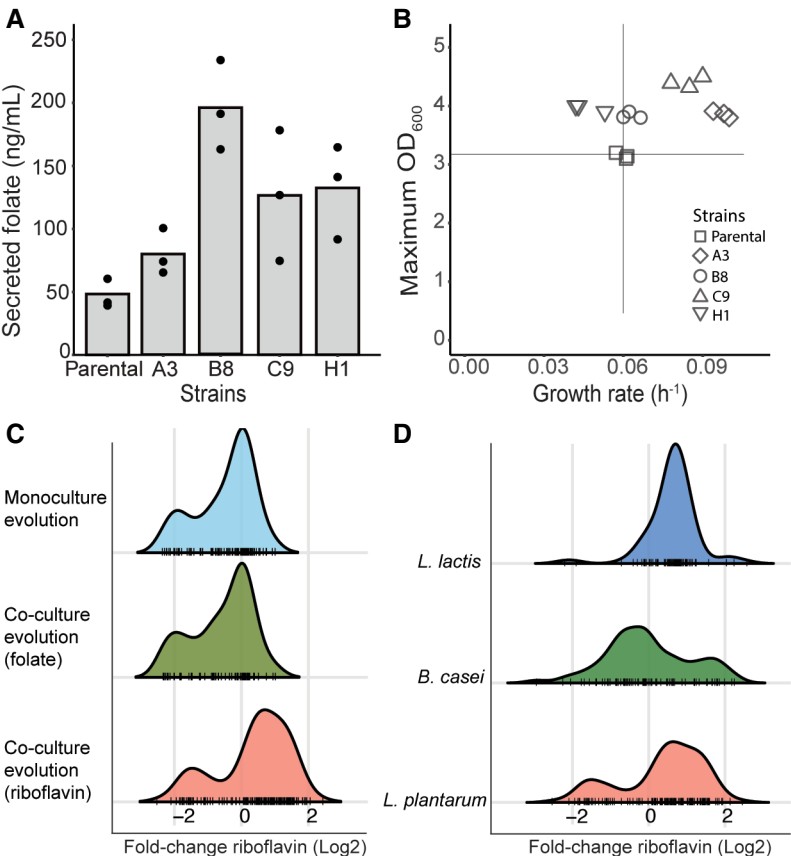

**Figure 3. Coevolution creates targeted selection pressure on exchanged metabolite.**

A Folate secretion by the parental *L. plantarum* and four evolved isolates. Bacteria were grown in amino acid supplemented medium and folate levels estimated using UPLC. *n* = 3 biological replicates.

B The growth rates and the maximum optical density reached by the parental and the evolved *L. plantarum* isolates cultured in amino acid supplemented medium. The vertical and the horizontal lines mark the average growth rate and optical density of the parental strain, respectively. *n* = 3 biological replicates.

C Riboflavin secretion by evolved isolates in comparison with the parental *L. plantarum* strain. The fold change is estimated based on fluorescence assay (Materials and Methods). Top: bacteria evolved in monocultures (*n* = 100 isolates). Middle: bacteria evolved in co-culture with folate auxotroph yeast (*n* = 80). Bottom: bacteria evolved in co-culture with riboflavin auxotroph yeast (*n* = 134). Black lines on abscissa mark data points for individual isolates.

D Riboflavin secretion by evolved isolates of three different bacterial species in comparison with their respective parental strain. The fold change is estimated based on fluorescence assay (Methods). Top: *L. lactis* evolved in co-culture with riboflavin auxotroph yeast (*n* = 97 isolates). Middle: *B. casei* evolved in co-culture with riboflavin auxotroph yeast (*n* = 77). Bottom: *L. plantarum* evolved in co-culture with riboflavin auxotroph yeast (*n* = 134). Black lines on abscissa mark data points for individual isolates.

Source data are available online for this figure.

their partner and under nutrient-rich conditions relevant for biotechnological applications.

# Discussion

Our results demonstrate that natural selection can be used to improve production of a desired compound by coupling it with the community fitness through cross-feeding. The link between the selection pressure exerted by the cross-feeding and the improved secretion phenotype was evident in the lack of enhancement in the monoculture evolution experiments. At the same time, the absence of riboflavin from the environment or genetic predisposition of the selected parental strain does not lead to increased riboflavin

secretion. The selection was also selective to the target vitamin, even when the two compounds used here, riboflavin and folate, originate from closely related pathways.

Mechanistically, genomic, proteomic, flux balance and metabolomic analyses revealed concordant changes in the regulation of vitamin biosynthesis and precursor supply (Fig 4). The increased abundances of the riboflavin and the purine biosynthetic enzymes, as well as the accumulation of glutamine and glycine, are in accord with the previous genetic engineering studies in *L. lactis* and *Bacillus subtilis* (Burgess *et al*, 2004; Schwechheimer *et al*, 2016). The relatively small number of altered protein and metabolite abundances further shows the targeted nature of the selection pressure. The focussed nature of these changes is important also from an application perspective; the evolved isolates are therefore likely not

to have alterations in other traits relevant for biotechnological processes.

In a mutualistic community, growth of all members is directly linked with that of the entire community. The phenotypes which co-operate more also receive higher rewards from their partners, an observation which has also been made for coevolution of auxo-trophic *Escherichia coli* strains (Zhang & Reed, 2014). Elsewhere, coevolution of mutualistic *E. coli* and *Salmonella enterica* resulted

into the majority of *S. enterica* cells exhibiting a co-operating pheno-type (Harcombe, 2010), while the growth rate of the co-cultures improved in expense of the individual growth rate of co-operating *E. coli* (Harcombe *et al*, 2018). In our case, the evolved bacterial isolates with enhanced riboflavin production not only exhibited improved growth but also supported a larger community popula-tion. The increased cost of riboflavin production was thus more than compensated for, through increased supply and more efficient

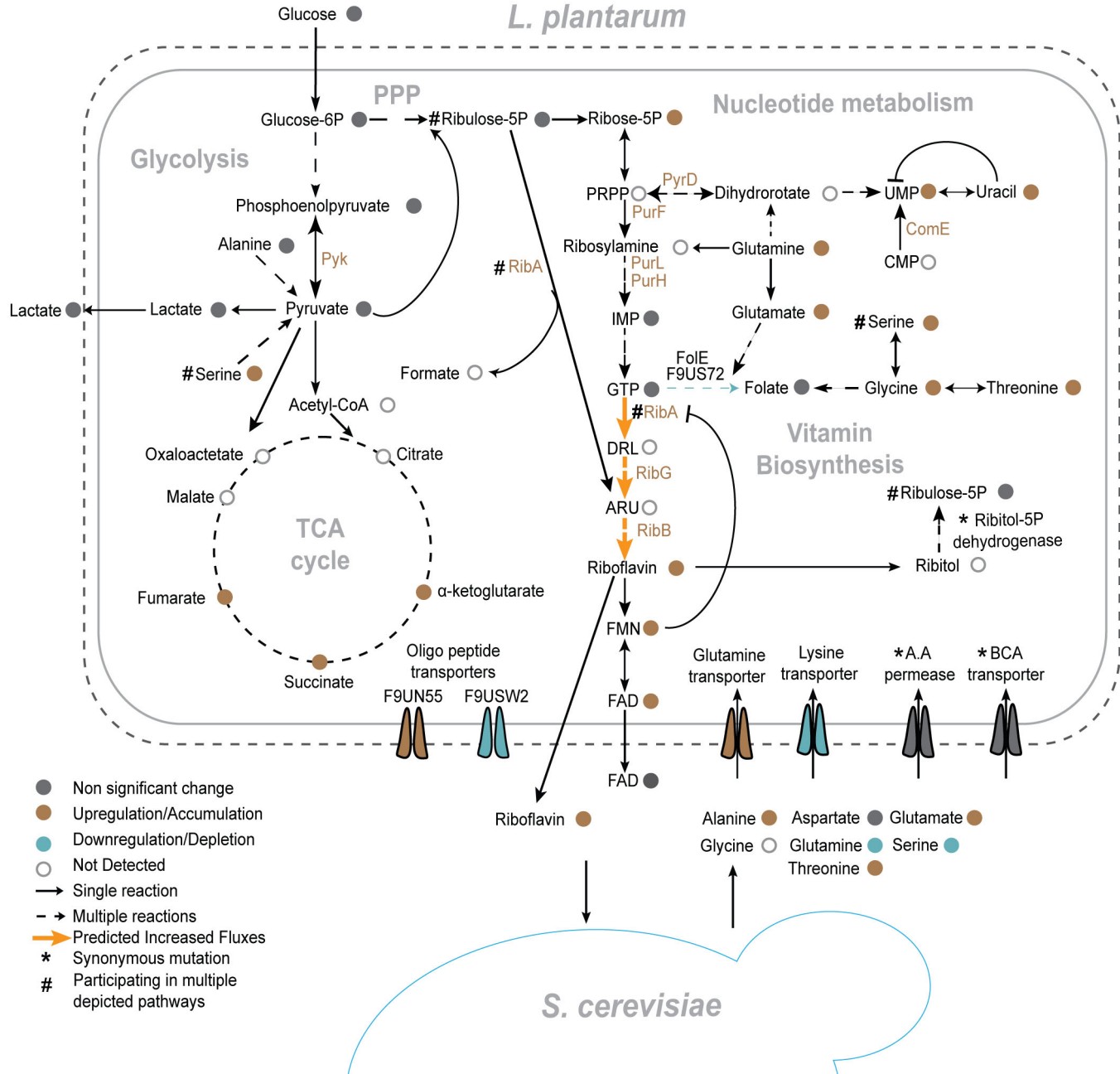

**Figure 4.  Key metabolic changes in *L. plantarum* following coevolution with yeast.**

Bacterial pathways associated with the biosynthesis of riboflavin and its main precursors, and amino acids secreted by yeast are shown. Metabolites and proteins with at least a twofold change in abundance and FDR-adjusted *P* < 0.05 are highlighted.

utilization of amino acids secreted by the yeast. Additionally, we show that the selection pressure for increased production is linked to the presence of the mutualistic partner and not with the absence of the compound from the environment. Our results, together with previous observations of amino acid exchange (Mee *et al*, 2014), demonstrate that increased secretion of costly metabolites is an exploitable strategy for adaptive laboratory evolution.

Bacterial cells with decreased or limited secretion capabilities could constitute cheaters that can lead the whole population to collapse (Hardin, 1968; Rankin *et al*, 2007; Jones *et al*, 2015). Yet, the majority of the isolates from the evolved communities exhibited improved secretion, and only 1 of 48 communities collapsed. These observations indicate that poor secretors did not have a sufficient fitness advantage over high riboflavin secretors. As expected for an obligate mutualism, and illustrated by our simulations (Fig EV4), improved vitamin secretion by bacteria results in positive feedback through increased availability of amino acids from yeast. This explains the amplified advantage to the co-operators, but not how "common goods"—riboflavin and amino acids—are not more frequently exploited by the cheaters. One likely explanation is the experimental conditions that we chose. All our evolution experiments were performed without shaking, and, as we show, the bacterial and yeast cells tend to form aggregates. Indeed, spatial segregation is one of the commonly suggested mechanisms of cheater suppression and mutualistic interaction stabilization (Momeni *et al*, 2013; Pande *et al*, 2016; Marchal *et al*, 2017). By affecting the distribution and availability of resources, spatial segregation allows preferential benefit to the co-operating partners (Dal Mitri *et al*, 2011; Hillesland, 2018; Co *et al*, 2020). In agreement, our simulations showed that increased mixing strength negatively affects the percentage of stronger mutualists (Fig EV4). Together, our experimental and simulation results show that the coevolution is robust against the emergence of non-co-operating phenotypes, while the fitness burden of increased vitamin production is balanced by evolution of better nutrient utilization. The high secretors thus remained prevalent.

Bacterial strains that produce either riboflavin or folate can be used to produce functional food with increased vitamin levels (LeBlanc *et al*, 2020). As our evolved bacterial isolates are not genetically engineered and retain their secreting phenotype in the absence of their yeast partner as well as when cultured in milk, they can be readily used in biotechnological applications to address nutritional deficiencies (Carrizo *et al*, 2020) or to treat patients with inflammatory gut diseases (von Martels *et al*, 2020). More broadly, our coevolution approach enables targeted improvement of fitness-costly metabolites through the operational simplicity offered by natural selection in laboratory evolution. The method can be readily applied to organisms wherein genetic engineering is yet not possible or restricted, and also in cases where the genetic basis of the trait of interest is complex and/or unknown. Beyond its biotechnological potential, the coevolution approach will also be valuable in addressing fundamental ecological questions on the emergence of cross-feeding and mutualism in microbial communities.

# Materials and Methods

## Reagents and Tools table

| Reagent/Resource | Reference or Source | Identifier or Catalog Number |
|---|---|---|
| **Experimental Models** | | |
| *Lactobacillus plantarum* | K.R. Patil Laboratory-Blasche S. | LPS |
| S90, *Saccharomyces cerevisiae* (MATα GAL2) | Steinmetz *et al*, 2002 | S90 |
| *Lactococcus lactis* | Blasche *et al* (2021) | 17S |
| *Brevibacterium casei* | Blasche *et al* (2021) | 303S |
| Δ*rib4rib5* (S90, *rib4rib5:kanMX4:Hyg*) | This study | DK1 |
| Δ*rib4rib5*RFP (S90, *rib4rib5:kanMX4:Hyg*, rfp:Clonat) | This study | DK2 |
| Δ*abz1* (*S90:Hyg*) | This study | DK3 |
| **Recombinant DNA** | | |
| puG6 | Euroscarf | Cat #P30114 |
| paG32 | Addgene | Cat #35122 |
| pCfB2049 | Stovicek *et al* (2015) | |
| **Oligonucleotides** | | |
| PCR primers | This study | Appendix Table S5 |
| **Software** | | |
| Breseq | https://barricklab.org/twiki/pub/Lab/ToolsBacterialGenome Resequencing/documentation/index.html Deathrage and Barrick (2014) | N/A |
| PEAR | https://cme.h-its.org/exelixis/web/software/pear/ | N/A |

                                                   

**Reagents and Tools table**   (continued)

| Reagent/Resource | Reference or Source | Identifier or Catalog Number |
|---|---|---|
| | Zhang *et al* (2014) | |
| SPAdes (version 3.15.1) | https://cab.spbu.ru/software/spades/ Nurk *et al* (2013) | N/A |
| RAST | https://rast.nmpdr.org Overbeek *et al* (2014) | N/A |
| Netlogo | https://ccl.northwestern.edu/netlogo/ Wilensky (1999) | N/A |
| Matlab (version 2018a) | https://www.mathworks.com/products/matlab.html | N/A |
| R A Language for Data Analysis and Graphics (version 3.6.3) | https://www.r-project.org | N/A |
| ImageJ | https://imagej.nih.gov/ij/ Schindelin *et al* (2012) | N/A |
| Konstantinidis_ALE_loop.nlogo | This paper | N/A |
| **Other** | | |
| 1290 Infinity II LC system | Agilent | N/A |
| Exion LC system | Sciex | N/A |
| Shimadzu TQ8040 GC | Shimadzu | N/A |
| Synergy HTX | Biotech | N/A |
| ACQUITY UPLC/UHPLC System | Waters | N/A |
| GeneVac EZ-2 plus evaporating system | SP Scientific | N/A |
| Riboflavin | Sigma-Aldrich | Cat #R4500 |
| FAD | Sigma-Aldrich | Cat # F6625 |
| FMN | Sigma-Aldrich | Cat #F2253 |
| Folate | Sigma-Aldrich | Cat #F7876 |
| Amino Acids | Sigma-Aldrich | Appendix Table S6 |

## Methods and Protocols

### Strains, media and growth conditions

The selected natural isolates of *Lactobacillus plantarum*, *Lactococcus lactis*, *Brevibacterium casei* and the prototrophic *Saccharomyces cerevisiae* (S90 strain) were grown as liquid cultures in MRS (De Man, Rogosa and Sharpe), GM17 (M17 Medium supplemented with glucose), MGAM (Gifu Anaerobic Medium Broth) and YPAD (Yeast extract, Peptone, Adenine, Glucose) respectively. Bacterial cultures were grown at 30°C without shaking, while the yeast cells were cultured at 30°C on an orbital shaker set to 180 rpm. Auxotrophic yeast cultures were supplemented with either 1 μg/ml riboflavin or 1 μg/ml folate. Pre-cultures were prepared in triplicated by cultivating a single isolated colony from a selective plate in the respective rich medium until mid-exponential phase (~18 h). Pre-cultured cells were centrifuged, washed three times with 1× PBS and starved for 3 h prior to the start of the experiments. Unless stated otherwise, all cultures were inoculated with an initial $OD_{600}$ of 0.01.

The chemically defined medium (CDM35, Ponomarova *et al*, 2017) was designed by reducing the number of components in a rich medium composed as the hybrid of previously described media, enabling co-culture of yeast and LAB strains. Two different versions of the CDM35 medium were used to create the proper evolution niches, CDM34R- (without riboflavin) and CDM33F- (where both

folate and para-aminobenzoic acid were omitted). Single bacteria were cultured in chemically defined medium supplemented with 40 ml/l of amino acid solution (CDM47) (Appendix Table S6). Two dropout versions of this media were also prepared CDM46R- (lacking riboflavin) and CDM45F- (without folate and para-aminobenzoic acid).

### Engineering of yeast auxotrophic strains

Riboflavin auxotrophy in the yeast strain S90 (Steinmetz *et al*, 2002) was designed by deleting two genes from the riboflavin biosynthetic pathway. Firstly, gene *RIB4* which encodes for the protein lumazine synthase was deleted, as lumazine is an immediate precursor of riboflavin. Positive clones were selected, and a second round of engineering was performed where *RIB5 gene was deleted. RIB5* encodes for riboflavin synthase, an enzyme that catalyses the last step of the riboflavin biosynthetic pathway. To generate a folate auxotrophic yeast strain (S90), *ABZ1* gene, encoding para-aminobenzoic synthase, was deleted. Gene deletions were performed by complete replacement of CDS of target genes with a dominant selection cassette by homologous recombination.

To double delete *RIB4* and *RIB5,* kanMX4 or hphMX4 cassettes were amplified by PCR from pUG6 (Güldener *et al*, 1996) and pUG32 (Goldstein & McCusker, 1999) (Appendix Table S5) using hybrid primers targeting the cassette and the locus flanking *RIB4* or

*RIB5*. Obtained PCR products were used to sequentially transform wild-type *S. cerevisiae* S90. For the single deletion of *ABZ1*, dominant selection cassette (kanMX4) was amplified from the *ABZ1* locus of the Δ*abz1* strain from the prototrophic *S. cerevisiae* gene deletion collection (Mûlleder *et al*, 2012) (Appendix Table S5). The PCR fragment was used to transform wild-type *S. cerevisiae* S90. For the transformation, yeast cultures in mid-exponential phase ($OD_{600} = 0.7$, 50 ml) were centrifuged, washed and re-suspended in 1 ml of sterile water. Then, 100 μl of the cell suspension was combined with an appropriate volume of transformation mix, which consisted of 240 μl PEG 3500 (50% w/v), 36 μl lithium acetate (1.0 M), 50 μl boiled single-stranded carrier DNA (2 mg/ml), 20 μl of PCR amplification product and 74 μl of ddH$_2$O. The cells with the transformation mixture were heat-shocked for 40 min at 42°C in a water bath, and then, they were re-suspended in YPAD and incubated for 3–4 h to allow for the expression of the integrated antibiotic marker. Finally, the cells were platted on YPAD agar plates containing the appropriate antibiotic (G418 300 μg/ml, hygromycin 150 μg/ml) supplemented with either riboflavin or folate and single colonies were selected. The success of homologous recombination was verified by colony PCR.

For genome integration of the constitutively expressed RFP in the riboflavin auxotrophic yeast strain, we used the pCfB2049 vector (Stovicek *et al*, 2015). The same transformation protocol was used, and positive mutants were selected in solid YPAD supplemented with riboflavin and nourseothricin sulphate (clonNat, 100 μg/ml). All the plasmids and the primers used in this study are described in Appendix Table S5.

### Adaptive laboratory evolution

In the present study, four similar mutualistic communities were constructed, each containing either different species of bacteria or different engineered yeast strains: (i) Δ*rib4:rib5 S. cerevisiae* and *L. plantarum* for increased production of riboflavin, (ii) Δ*abz1 S. cerevisiae* and *L. plantarum* for increased production of folate, (iii) Δ*rib4:rib5 S. cerevisiae* and *L. lactis* and (iv) Δ*rib4:rib5 S. cerevisiae* and *B. casei* for production of riboflavin. At the beginning of the evolution experiment, single bacterial colonies were grown as described previously, were washed and mixed with the appropriate auxotrophic yeast strain. In total, 12 co-cultures of each community were started in parallel, with each containing both species in 1:1 volume ratio based on their $OD_{600}$. The initial $OD_{600}$ value of the communities was equal to 0.01, and they were inoculated in 96-well plates containing 200 μl CDM34R- or CDM33F-. The evolution was performed under microaerobic conditions (statically) at 30°C. Growth was monitored in a Biotech microplate reader, and when the communities reached stationary phase, the appropriate volume from each well was transferred to fresh medium with a starting $OD_{600}$ value of 0.01. Stability of the communities was estimated by platting on YPAD plates, where all members could grow and form colonies. Intermediate and final populations were stored by adding 80% [w/v] glycerol directly to the plate wells (final concentration 30% [w/v]) and then stored at −80°C. The total duration of the experiment and the number of generations varied between species and between communities. *L. plantarum* communities were evolved for riboflavin production for 3 months, and during this period, the populations were transferred 24 times, while a total number of 160 generations passed. Estimation of the number of generations is

based on the measured $OD_{600}$, using the following formula $Log_{10}(A_f/A_i)]/0.3$, where $A_f$ is the $OD_{600}$ before the transfer and $A_i$ is the $OD_{600}$ that was initially inoculated. The evolution of *L. plantarum* for folate production lasted 2 months, with a total of 15 transfers and 80 generations. Evolution of *B. casei* and *L. lactis* lasted for a total of 80 days and approximately 140 generations in the case of riboflavin, and 80 generations in the case of folate, passed.

For the ALE monoculture experiment of *L. plantarum* in CDM46R-, we followed the same procedure described above with the difference that we started with 4 replicates in parallel. This evolution experiment lasted for 2 months and in total 100 generations passed.

### Estimation of riboflavin secretion through fluorescence intensity assay

All final populations from each community were plated on solid media, each composed of the appropriate selective rich medium and 2% [w/v] agar as solidifying agent. From each population, at least seven single colonies were picked and transferred to a separate well of a 96-well plate filled with 200 μl of CDM46R-. The isolates were assigned a letter and a number that signified their position in the 96-well plate in which they were isolated. The communities were numbered (1–12), and each single colony isolate was identified by a letter (A - G). For example, strain A1 denotes isolate "A" from community 1. Since one of the communities (community 9) collapsed during the adaptive laboratory evolution experiment, the twelfth line of the plate was filled with random isolates from all the other communities to increase the number of screened isolates. In a similar fashion 4 single colonies of the corresponding parental strain were isolated and inoculated in the same 96-well plate, while a well with only CDM46R- was used as an empty control. The plate was incubated for 72 h, which was on average the time point between the passages during the serial transfer experiment. After the 72 h, the plate was centrifuged for 3 min at 200 *g* in an Eppendorf centrifuge and the top 100 μl of supernatant was transferred to a black opaque 96-well plate for fluorescence measurement. The measurements were performed in a Biotech plate reader, using the 440-nm excitation at medium intensity and detecting at 520 nm. The average fluorescence intensity for the parental strain was calculated, and this value was used to calculate the log$_2$fold change of the evolved isolates.

### Determination of riboflavin secretion in a complex matrix: extraction from pasteurized milk samples

The ability of the bacterial monoculture to retain the riboflavin secreting phenotype growing in milk was determined for the parental and two evolved *L. plantarum* isolates (B4 and E6). All bacterial isolates were inoculated in 4 ml of pasteurized milk with an initial $OD_{600}$ of 0.01 and cultured statically at 30°C for a total of 72 h. Total riboflavin was extracted from 0.5 ml of culture with the hot acid extraction method, based on Panfili *et al* (1995) with some modifications. In short, 0.5 ml of culture was transferred in a glass vial and 50 μl of 1 M HCl was added on top. Then, the samples were autoclaved at 120°C for 30 min, after which were left to cool down at RT. Next, 750 μl of 1 M sodium acetate was added to the autoclaved samples to bring the pH close to 4.5. The total volume of each sample was transferred to an Eppendorf tube and was then centrifuged for 15 min at 20,000 *g* in an Eppendorf centrifuge. The

supernatant was filtered through a 0.22-μm syringe filter into a new Eppendorf tube and stored at −80°C until further analysis with liquid chromatography.

### Liquid chromatography–mass spectrometry (LC-MS) sample preparation and data analysis for the detection of vitamins and nucleotide-related metabolites

#### Sample preparation

Riboflavin was extracted from 2 ml of culture supernatant by addition of an equal amount of 80:20 $H_2O$: ACN. The samples were left at 4°C for 10 min and then were transferred to dark glass vials. Next, they were concentrated using a vacuum concentrator, at 30°C for 2 h and subsequently reconstituted in 200 μl of 80:20 $H_2O$: ACN. The reconstituted samples were filtered through a 0.22-μm syringe filter.

#### LC-MS measurements

The metabolite extracts were separated using HILIC on a 2.1 × 100 mm, 3.5 μm BEH amide column (Waters). Separation was achieved by gradient elution (100% B to 100% A in 20 min) on a binary solvent Exion LC system (Sciex, Darmstadt, Germany). Mobile phase A consisted of 80:20 $H_2O$: ACN, and mobile phase B consisted of 85:15 ACN: $H_2O$. Both mobile phases A and B were modified with 10 mM ammonium acetate and adjusted to pH 8.5 with 25% [v/v] ammonium hydroxide. A flow rate of 300 μl / min was used for separation, and the column and sample tray were held at 25 and 4°C, respectively. A 5 μl injection volume was used for all samples.

#### MS analysis

MS analysis was performed in the MRM-scan mode on a Q-Trap 6500+ (Sciex, Darmstadt, Germany) equipped with an Ion Drive Turbo V electrospray ionization probe. In both the positive and negative ionization modes, the source temperature was set to 550°C, the curtain gas was set to 30, the nebulizer gas (GS1) and the auxiliary gas (GS2) were both set to 45, and the collision activation dissociation (CAD) gas was set to medium. Nitrogen was used as both the curtain gas and the CAD gas. In the positive ionization mode, the spray voltage was set to 5.5 kV, the declustering potential to 90, and the entrance and exit potentials to 10. In the negative ionization mode, the spray voltage was set to −4.5 kV, and the declustering potential to −60 and the entrance and exit potentials were both set to −10. Both the LC and the MS were controlled using Analyst software version 1.7.

The MRM transitions used for analysis of the vitamins were as follows: Folate 442/295.1, Riboflavin 377/243 and FAD 786/348 all analysed in the positive ionization mode, and, for FMN 455/213 analysed in the negative mode. The MRM transitions used for the analysis of the nucleotides were as follows: IMP 349/137, UMP 325/97, GMP 364/154, AMP 348/136 all in the positive ionization mode.

Data analysis was performed using the Sciex OS software. A Gaussian smoothing of 1 point was applied to all peaks.

### Liquid chromatography–mass spectrometry sample preparation and data analysis for the detection of amino acids

#### Sample preparation

Liquid samples were prepared for LC-MS and LC-MS/MS analyses through the addition of an equal volume of ice-cold (90:10) ACN:

5 mM ammonium acetate (pH 9). Extractions were incubated for 1 h at −20°C followed by centrifugation at 2,000 g for 10 min at 4°C. 20 μl of extraction supernatant was transferred to Nunc 96-well, V-shape plates, closed with temperature-sensitive seals and stored at −80°C until further analysis.

#### LC-MS and LC-MS/MS measurements

Samples were prepared as described above and analysed as previously reported (Agilent, 2019). In brief, chromatographic separation was achieved using Agilent InfinityLab Poroshell 120 HILIC-Z column, 2.1 mm × 150 mm, 2.7 μm column and an Agilent 1290 Infinity II LC system coupled to a 6550 iFunnel qToF mass spectrometer. Column temperature was maintained at 45°C, and the following mobile phase was used: mobile phase A: ammonium acetate 5 mM, pH 9 and 250 μM InfinityLab Deactivator and mobile phase B: ammonium acetate in ACN: $H_2O$ 85:15 (v/v) 5 mM, pH 9 and 250 μM InfinityLab Deactivator. 5 μl of sample was injected at 96% mobile phase B, maintained for 2 min, followed by linear gradient up to 88% B in 3.5 min, maintained for 3 min, followed by a linear gradient to 86% B in 0.5 min and maintained at 86% for 5 min, then linear gradient to 82% mobile phase B over 3 min, and a linear gradient to 65% B over 5 min, which was maintained for 1 min. The column was allowed to re-equilibrate HILIC conditions for 8 min before each sample injection. The Agilent LC-MS 6550 qTOF instrument was operated in negative scanning mode (60–1,600 $m/z$) with the following source parameters: VCap, 3,500 V; nozzle voltage, 0 V; gas temperature, 225°C; drying gas 13 l/min; nebulizer, 35 psi; sheath gas temperature 350°C; sheath gas flow 12 l/min, fragmentor, 125 V and skimmer, 45 V. Online mass calibration was performed using a second ionization source and a constant flow (10μl/min) of reference mass solution (119.0363 and 1033.9881 $m/z$).

#### LC-MS data analysis

The MassHunter Quantitative Analysis Software (Agilent Technologies, version 10.0) was used for both LC-MS and LC-MS/MS molecular feature extractions. The following setting was applied for feature extraction: peak filter of absolute height: 10,000 counts, limit assigned charge states to 1, only H-charged molecules were included and compound quality score should be greater than 80%. Peak alignment and identification were carried out using Mass Profiler Professional (Agilent, version 15.1) with default parameters: mass tolerance of 2 mDa or 20 ppm and retention time tolerance of 0.3 min or 2%. Extracted and aligned compounds were annotated using the Metlin PCDL B.08.0 metabolite and peptide database/library.

### Growth kinetics in CDM46R- and CDM46F-

Biological triplicates of the bacterial parental and the evolved isolates were inoculated in rich medium and were left to grow overnight. These overnight cultures were washed three times with 1X PBS and starved for 2 h. Then, cells were inoculated with an initial $OD_{600}$ of 0.01 in a final volume of 7 ml in 50-ml Erlenmeyer flasks and were grown statically at 30°C. Estimation of growth was based on increase in the optical density which was measured at regular time intervals. Growth was followed until the $OD_{600}$ remained stable. Growth rates were estimated by calculating the maximum slope values of the best fit to log2-transformed OD values. At least 5

points were used for these calculations. Estimation of the growth curve in CDM46R- with diluted amino acids was performed in a similar fashion. In this experiment, three biological replicates of the parental and the selected evolved isolates were inoculated in 200 μl of CDM46R- in a 96-well plate. CDM46R- was supplemented both with 40 ml/l and with 5 ml/l of amino acid solution (Appendix Table S6), in technical replicates. The 96-well plate was sealed with a Breathe-Easy® sealing membrane (Z380059, Sigma-Aldrich), and growth was followed on a Biotech microplate reader at regular time intervals.

### Conditioned media assays
#### Bacterial conditioned medium
Bacterial isolates were cultured in CDM46R- medium with an initial inoculation $OD_{600}$ of 0.01, until mid-exponential phase, the cultures were centrifuged for 3 min at full speed, and then, the supernatant was filtered through a 0.22-μm syringe filter. Auxotrophic yeast cells from an overnight culture were washed three times in 1× PBS, starved for 3 h and then inoculated in the bacterial conditioned medium without any further supplementation. The initial $OD_{600}$ value was 0.01 in a final volume of 200 μl in a 96-well plate, which was sealed with a Breathe-Easy® sealing membrane. Growth was monitored in a Biotech microplate reader, with $OD_{600}$ measurements every hour for a total of 48 h. After the cultures had reached stationary phase, the 96-well plate was thoroughly mixed and the $OD_{600}$ was measured one last time. Growth rates were estimated by calculating the maximum slope values of the best fit to log2-transformed OD values. Between 6 and 8 points were used for these calculations.

#### Yeast conditioned medium
Wild-type *S. cerevisiae* S90 cells were cultured in CDM34R- with an initial inoculation $OD_{600}$ of 0.01, until they reached mid-exponential phase. Then, the cultures were centrifuged at full speed for 3 min and the medium supernatant was filtered with 0.22-μm syringe filters. Bacterial cells (parental and evolved) from overnight cultures were washed three times in 1× PBS and starved for 2 h. Then, they were inoculated in yeast conditioned medium with an initial $OD_{600}$ of 0.01 in a final volume of 200 μl in a 96-well plate, which was sealed with a Breathe-Easy® sealing membrane. All bacterial isolates were inoculated in conditioned medium originating from the same yeast culture.

### Flow cytometry
Auxotrophic RFP expressing yeast cells and bacterial cells (parental and evolved) were inoculated in CDM34R- with the same initial $OD_{600}$ of 0.01. At specific time points, a total volume equal to 0.5 ml of *L. plantarum* and *S. cerevisiae* co-culture was pelleted. The pellet was fixed with 4% [w/v] formaldehyde-PBS solution at RT for 10 min. The fixed pellet was washed once with 1× PBS and then rehydrated in phosphate-buffered saline with 1 mM EDTA. Prior to flow cytometry, the pellets were briefly sonicated three times (10 s with 0.5-s ON-OFF intervals; 10% amplitude; Branson Sonifier W-250 D, Heinemann) at 4°C. Samples were analysed by flow cytometry using a Cytek Aurora Cytometer (Cytek) at low flow. A total volume of 35 μl was analysed for each sample, and thresholds used were 800 for forward scatter height and 600 for side scatter height. Yeast cells were identified based on the RFP intensity, and cells below the intensity threshold of $10^4$ were excluded from the analysis.

### Microscopy
Brightfield and fluorescence images were acquired at the Advanced Light Microscopy Facility at EMBL, Heidelberg with a ZEISS Cell Observer Widefield microscope, equipped with a metal halide fluorescence lamp. Observations were made using a 63× oil objective, and the acquired images were analysed with ImageJ (Schindelin *et al*, 2012).

### Cell surface traits
#### Autoaggregation and co-aggregation
Evaluation of the cell aggregation phenotype was performed as described previously (Kotzamanidis *et al*, 2010). In short, overnight cultures of bacteria and WT S90 yeast in biological triplicates were washed three times in 1× PBS. The $OD_{600}$ was measured, and an appropriate volume of the washed cultures was transferred to a cuvette to have a starting $OD_{600}$ of 1. For the co-aggregation experiment, bacteria and yeast were mixed with a 4:1 ratio (80% bacterial cells and 20% yeast cells) to better represent the higher number of bacterial cells that were detected during co-culture experiments. All the cuvettes from both experiments were left still in a place with no vibrations for the duration of the experiment, and the $OD_{600}$ of each one was measured every hour. A final measurement was performed after 24 h. The percentage of aggregation is calculated based on the decrease in the measured $OD_{600}$ values, using the formula (1- $A_t$ / $A_0$) × 100, where $A_t$ is the $OD_{600}$ value of every time point and $A_0$ is the starting $OD_{600}$ value.

#### Microtiter dish biofilm formation assay
Assessment of the biofilm formation ability was performed for both bacterial monocultures and bacteria–yeast co-cultures as described by Merritt *et al* (2005) with modifications. In short, overnight cultures were diluted in either CDM46R- (bacterial monoculture) or CDM35R- (co-cultures) to an $OD_{600}$ of 0.02 and 100 μl aliquots were transferred to 96-well polystyrene dishes. For each mono- or co-culture, we used 4 replicates with the ratio between bacteria and yeast in the co-culture being 1:1 based on their OD. The dishes were incubated at RT for 48 h, after which media and planktonic cells were removed and the wells were washed four times with ddH₂O. The adherent biofilm was stained with 200 μl of 0.01% [w/v] crystal violet for 15 min, and subsequently, the wells were washed four times with ddH₂O. Bound dye was solubilized with 200 μl of 80% ethanol–20% acetone, and the $A_{570}$ was measured in a Biotech plate reader. Background absorbance for empty control wells was subtracted, and the mean and standard deviation (SD) were calculated from the replicates.

### Genomic DNA extraction from lactic acid bacteria
Overnight bacterial cultures of 4 ml total volume were pelleted in 2-ml Eppendorf tubes and were re-suspended in 1 ml of TES buffer (Tris50 mM, EDTA 20 mM and sucrose 2 mM) which contained 20 mg/ml of lysozyme. The tubes were incubated for 30 min at 37°C, after which the cell suspensions were transferred to FastPrep Cap tubes with 200 μl of glass beads (150–212 nm acid washed, Sigma) and the cells were mechanically disrupted with three rounds of bead beating at 4.5 Mhz/s for 20 s with 1-min cooling intervals. Then, 150 μl of 20% [w/v] SDS was added to the tubes and they were incubated for 5 min at RT, before being centrifuged for 1 min at full speed. The supernatant was collected in a fresh

tube, and 100 µl of proteinase K (20 mg/ml) was added, and the samples were incubated for 30 min at 37°C, followed by the addition of 200 µl of 5 M potassium acetate and incubation for 15 min on ice. The tubes were centrifuged for 15 min at max speed at 4°C, and the supernatant was transferred to a new tube in a 1:1 ratio with phenol–chloroform/isoamyl alcohol. Then, the tubes were centrifuged briefly until an emulsion was formed. The emulsions were centrifuged at 16,000 *g* for 10 min at RT, the aqueous phase was transferred to a new tube and mixed with an equal volume of ice-cold 100% ethanol, and DNA was precipitated overnight at −20°C. Next day the tubes were centrifuged for 10 min at max speed at 4°C, and the DNA pellets were washed with 70% [v/v] ethanol and then dried for 20 min at 30°C. The dried pellets were dissolved in 60 µl of ddH$_2$O with 100 µg/ml RNAse and were incubated at 37°C for 15 min. The quality of the extracted DNA was evaluated by agarose gel electrophoresis (1% [w/v]). DNA concentrations were measured using a Qubit (Thermo Fisher Scientific, USA). Dissolved DNA samples were stored at 4°C until further processing.

### Whole genome sequencing and data analysis

Equal amounts of DNA from all samples were used for library preparation, which was done with the NEBNext DNA Ultra2 Library Preparation Kit (New England Biolabs). The preparation of the library was performed on an automated liquid handling system (Hamilton Robotics), the quality of the library was tested on a 2100 BioAnalyzer (Agilent Technologies), and the DNA concentration was measured using a Qubit. Sequencing was performed at the Genomics Core Facility (EMBL, Heidelberg) with the use of the Hiseq2500 platform (Illumina, San Diego, USA) and the run produced 250 bp paired-end reads. Identification of mutations that occurred during the ALE co-culture experiment was performed with Breseq (Deathrage & Barrick, 2014). To compare the sequence of the parental strain with evolved isolates, we initially assembled the parental sequence. The initial quality of the raw Illumina paired-end reads was evaluated with FastQC (Babraham Bioinformatics) and was subsequently trimmed with CUTADAPT (Martin, 2011) setting a quality threshold of 15 and a length cutoff of 150 bp. Next, the pair-ended files were merged using PEAR (Zhang *et al*, 2014) with the default parameters. Assembly of the connected reads was done with SPAdes (Nurk *et al*, 2013) using the parameters --careful, -k 5, 15, 35, 55 and the rest as default, while the evaluation of the assembly was done with QUAST (Gurevich *et al*, 2013). The assembled genome was annotated with RAST (Overbeek *et al*, 2014) through which we identified as the closest related *L. plantarum* strain to ours the strain LP3, for which the whole genome sequence was available in NCBI (direct submission). We used this strain as a reference sequence in Breseq, as its inputs are GFF3 files.

We used Breseq to compare the genome of strain LP3 and the genome of the parental strain, with the default parameters and additionally the commands –l 120, -j 4. As it was expected, we were able to identify differences between our parental and strain LP3, which we then applied to the reference genome, with the gdtools' command APPLY, which is provided by Breseq as well. This way we created a "mutated" reference genome which was used for all further comparisons with the evolved isolates, while keeping the same Breseq parameters, as described for the parental strain.

### Protein sample preparation, sequencing and analysis

Biological triplicates of the parental and the evolved bacterial isolates were inoculated in CDM46R- with an initial OD$_{600}$ of 0.01. For the extraction of total proteome, 10 ml of each culture was transferred into ice-cold 15-ml Falcon® tubes which were centrifuged immediately at 850 *g* for 3 min at 4°C (Eppendorf centrifuge). The supernatant from the centrifugation was discarded, and the cell pellets were washed once with 1 ml of cold PBS buffer. The washed pellets were snapped frozen with liquid nitrogen and stored at −80°C. For the extraction, the cell pellets were lysed with 0.1% RapiGest (Waters) in 100 mM ammonium bicarbonate, followed by mechanical disruption with three rounds of sonication (1 cycle: 10-s sonication and 10-s rest on ice per round). Sonication was followed by 2 cycles of bead beating (200-µl glass beads, 150–212 nm acid washed, Sigma), each cycle lasting 20 s at 4 Mz/s with 1-min cooling intervals between the cycles. Reduction in disulphide bridges in cysteine containing proteins was performed with dithiothreitol (56°C, 30 min, 10 mM in 50 mM HEPES, pH 8.5). Reduced cysteines were alkylated with 2-chloroacetamide (RT, in the dark, 30 min, 20 mM in 50 mM HEPES, pH 8.5). Samples were prepared using the SP3 protocol (Hughes *et al*, 2019), and trypsin (sequencing grade, Promega) was added in an enzyme to protein ratio 1:50 for overnight digestion at 37°C. Peptides were labelled TMT6plex (Dayon *et al*, 2008) Isobaric Label Reagent (ThermoFisher) according the manufacturer's instructions. For further sample clean up, an OASIS® HLB µElution Plate (Waters) was used. Offline high pH reverse phase fractionation was carried out on an Agilent 1200 Infinity high-performance liquid chromatography system, equipped with a Gemini C18 column (3 µm, 110 Å, 100 × 1.0 mm, Phenomenex, Reichel *et al*, 2016) resulting in 12 fractions. After fragmentation, the peptides were separated using an UltiMate 3000 RSLC nano LC system (Dionex) fitted with a trapping cartridge (µ-Precolumn C18 PepMap 100, 5 µm, 300 µm i.d. × 5 mm, 100 Å) and an analytical column (nanoEase™ M/Z HSS T3 column 75 µm × 250 mm C18, 1.8 µm, 100 Å, Waters). Trapping was carried out with a constant flow of trapping solution (0.05% trifluoroacetic acid in water) at 30 µl/min onto the trapping column for 6 min. Subsequently, peptides were eluted via the analytical column running solvent A (0.1% [v/v] formic acid in water) with a constant flow of 0.3 µl/min, with increasing percentage of solvent B (0.1% [v/v] formic acid in acetonitrile) from 2 to 4% in 4 min, from 4 to 8% in 2 min, then 8–28% for a further 37 min, in another 9 min from 28 to 40%, and finally 40–80% for 3 min followed by re-equilibration back to 2% B in 5 min. The outlet of the analytical column was coupled directly to an Orbitrap QExactive™ plus Mass Spectrometer (Thermo) using the Nanospray Flex™ ion source in positive ion mode. The peptides were introduced into the QExactive plus via a Pico-Tip Emitter 360 µm OD × 20 µm ID; 10 µm tip (New Objective) and an applied spray voltage of 2.2 kV. The capillary temperature was set at 275°C. Full mass scan was acquired with mass range 375–1,200 *m/z* in profile mode with resolution of 70,000. The filling time was set at maximum of 100 ms with a limitation of 3 × 10$^6$ ions. Data-dependent acquisition (DDA) was performed with the resolution of the Orbitrap set to 17,500, with a fill time of 50 ms and a limitation of 2 × 10$^5$ ions. A normalized collision energy of 32 was applied. Dynamic exclusion time of 20 s was used. The peptide match algorithm was set to "preferred" and charge exclusion "unassigned", charge states 1, 5–8 were excluded. MS data were acquired in profile mode. The

acquired data were processed using IsobarQuant (Franken *et al*, 2015) and Mascot (v2.2.07). A UniProt *L. plantarum* proteome database (UP000000432) containing common contaminants and reversed sequences was used, with the addition of the sequences of the absent riboflavin biosynthetic proteins (RibD and RibF). The search parameters were the following: carbamidomethyl (C) and TMT10 (K) (fixed modification), acetyl (N-term), oxidation (M) and TMT10 (N-term) (variable modifications). A mass error tolerance of 10 ppm was set for the full scan (MS1) and for MS/MS (MS2) spectra of 0.02 Da. Trypsin was selected as protease with an allowance of maximum two missed cleavages. A minimum peptide length of seven amino acids and at least two unique peptides were required for a protein identification. The false discovery rate on peptide and protein level was set to 0.01.

### Metabolic Analysis with Relative Gene Expression (MARGE)

The qualitative effect of metabolic enzyme abundance changes on riboflavin generation in the evolved isolates was predicted using genome-scale metabolic model simulations. The genome-scale metabolic model iBT721 of *L. plantarum* (Teusink *et al*, 2006) was obtained from https://github.com/opencobra/m_model_collec tion/blob/master/sbml3/iBT721_fixed.xml. The model did not contain a complete riboflavin synthesis pathway as the strain we used. Thus, the pathway was introduced into the model by adding GTP cyclohydrolase II (EC 3.5.4.25), diaminohydroxyphosphoribo-sylaminopyrimidine deaminase (EC 3.5.4.26), 5-amino-6-(5-phosphoribosylamino)uracil reductase (EC 1.1.1.193), 5-amino-6-(5-phospho-D-ribitylamino)uracil phosphatase (EC 3.1.3.104), 6,7-dimethyl-8-ribityllumazine synthase (EC 2.5.1.78) and riboflavin synthase (EC 2.5.1.9) reactions. A growth medium for the model was set up according to the experimental growth medium with the exchange flux bounds in the model set to represent the relative molar availabilities of the nutrients. A couple of nutrients (pyridoxamine and 2-aminoethylphosphonate) was additionally introduced to allow the model to grow. The model simulations were performed using Metabolic Analysis with Relative Gene Expression (MARGE) method, which belongs to the *reframed* python package (https://github.com/cdanielmachado/reframed). The experimentally determined growth rates of the parental strain and evolved isolates were set as the growth lower bounds for the simulations, and the total nutrient uptake excluding oxygen was used as the objective. For the parental strain, the lower bound of riboflavin production was set to the specific production rate of $6.13 \cdot 10^{-6}$ mmol/(g CDW h) estimated from experimental data. The specific production rate was estimated from the extracellular riboflavin titter after 72 h of cultivation and assuming a half of the cell concentration derived from $OD_{600}$ of 2 using the dependency by Adler *et al* (2013). The riboflavin production by the evolved isolates was left unconstrained and to be predicted from the metabolic enzyme abundancy changes by MARGE. Significant metabolic enzyme abundance changes with 50% or beyond fold changes ($\log_2$ fold change $\geq 0.5$), extracted from the proteomics analysis, were used as input for MARGE. MARGE was run by equal metabolic state optimality relaxation of 3% for both parental strain and evolved isolates. MARGE and *reframed* v. 1.2.0 were run using cplex v. 12.8.0.0 and Python v. 3.6.12 with miniconda v. 4.9.2. COBRApy v. 0.20.0 (Ebrahim *et al*, 2013) was used for the model manipulation.

### Gas chromatography–mass spectrometry (GC-MS) sample preparation, data acquisition and analysis

Analysis of extracellular and intracellular metabolites was performed on cultures that were sampled at mid-exponential phase, as described previously (Strucko *et al*, 2018). Briefly, 10 ml of culture was vacuum-filtered through nylon membrane filters (0.45 μm, Whatman™) and the filtered medium was immediately stored at −80°C until further analysis. For sample preparation, 50 μl of filtered medium was added to 300 μl of cold (−20°C) HPLC grade methanol (Biosolve Chimie, France)/Milli-Q water (1:1, [v/v]) and the samples were incubated at 72°C for 15 min, while ribitol (Adonitol) (Alfa Aesar, UK) was added as an internal standard. The metabolites were centrifuged at 9,400 *g* at 0°C for 10 min, and the supernatants were collected and dried with the Genevac EZ-2 Plus evaporator (program, hplc fraction; temperature, 30°C). The cell pellet on the membrane filters was washed in three rapid steps with 5 ml of room temperature PBS to ensure no contamination from extracellular metabolites. The polar metabolites were extracted by adding the washed cell-containing filter in 5 ml of cold (−20°C) HPLC grade methanol (Biosolve Chimie, France)/Milli-Q water (1:1, [v/v]) and incubating at −20°C for 1 h, followed by incubation at 72°C for 15 min, while ribitol (adonitol, Alfa Aesar, UK) was added as an internal standard. The mixture of metabolites and cell debris was centrifuged at 9,400 *g* at 0°C for 10 min, and the supernatants were collected and dried with the Genevac EZ-2 Plus evaporator (program, hplc fraction; temperature, 30°C). The dried polar metabolites were derivatized with 100 μl of 20 mg/ml methoxya-mine hydrochloride (Alfa Aesar, L08415.14) solution in pyridine (Alfa Aesar, A12005) for 90 min at 37°C, followed by reaction with 200 μl *N-methyl-N-395 (trimethylsilyl)trifluoroacetamide (Alfa Aesar, A13141)* for 10 h at RT as justified in Kanani and Klapa (2007). GC-MS analysis was performed using a Shimadzu TQ8040 GC-(triple quadrupole) MS system (Shimadzu Corp.) equipped with a 30 m × 0.25 mm × 0.25 μm ZB-50 capillary column (Phenom-enex, 7HG-G004-11). 1 μl of sample was injected in split mode (split ratio 1:40) at 250°C using helium as a carrier gas with a flow rate of 1 ml/min. GC oven temperature was held at 100°C for 4 min followed by an increase to 320°C with a rate of 10°C /min and a final constant temperature period at 320°C for 11 min. The interface and the ion source were held at 280 and 230°C, respectively. The detector was operated both in scanning mode recording in the range of 50–600 *m/z* and in multiple reaction monitoring (MRM) mode for specified metabolites. For peak annotation, the GCMSsolution software (Shimadzu Corp.) was utilized. The metabolite identification was based on an *in-house* database with analytical standards utilized to define the retention time, the mass spectrum and the quantifying ion fragment for each metabolite. The metabolite quantification was carried out by calculating the area under the curve (AUC) of the quantifying ion fragment of each metabolite divided by the AUC of ribitol's quantifying ion fragment 319 (*m/z*).

### Ultra-performance liquid chromatography (UPLC) sample preparation, data acquisition and analysis for the detection of folate

Folate was extracted from 2 ml of culture supernatant by addition of an equal amount of 20% AcN: H2O. The samples were left at 4°C for 10 min and then were transferred to dark glass vials. Next, they were concentrated using a vacuum concentrator, at 30°C for 2 h and

subsequently reconstituted in 200 µl of 20% AcN:H$_2$O. The reconstituted samples were filtered through a 0.22-µm syringe filter. The liquid chromatography method for the detection of folate was run on a Waters Acquity UPLC H-Class instrument with a PDA detector and a quaternary solvent system. The established method was 8 min long, with a flow rate of 0.35 ml/min and run on a BEH Hilic column (130 Å, 1.7 µm, 2.1 mm × 150 mm, Waters, part number 176001094). The column was heated to 35°C, samples were kept at 6°C, and a total volume of 5 µl was injected. The method used 50% [v/v] acetonitrile (CAN; Biosolve, ULC grade) as washing buffer and 50% [v/v] methanol (Biosolve, ULC grade) as purging buffer. The organic mobile phase was acetonitrile, 10 mM ammonium acetate (Sigma-Aldrich, HPLC grade) was used as buffer, and the pH was adjusted to 9. The two buffers used were 50% CAN + 10 mM ammonium acetate pH 9 and 100% CAN + 10 mM ammonium acetate pH 9. The full method is described in Appendix Table S7. The elution time of folate was estimated by using pure grade standards (≥ 97%, Sigma-Aldrich). All chromatograms were annotated with the vendor-specific program Empower 3, and manually curated for peak identification. The readout for all chromatograms is the baseline-corrected area under the curve of the vitamin.

### Model setup

The model is implemented with Netlogo (Wilensky, 1999), a freely available framework for the setup and simulation of agent-based models. The model world consists of a grid of 32 × 32 patches constituting the liquid growth medium. The agents (described in the next paragraph) secrete amino acids or vitamins into these patches. This means, each patch contains (owns) a certain amount of amino acids and vitamin. A diffusion parameter in the model determines how much of these metabolites diffuse out of the focal patch into the neighbouring patches in each time step.

The mobile agents (called turtles in Netlogo) in the model are yeast, bacteria and mutant bacterial cells. Individuals can move at random which simulates liquid culture mixing due to shaking. Each agent starts with a specified amount of energy. In the course of the simulation, this energy is consumed by the secretion of metabolites into the medium and replenished by consuming metabolites from the medium. A transformation efficiency determines how many units of energy are gained from one unit of consumed metabolite. A secretion cost determines how many units of energy are spend for the secretion of one unit secreted metabolite (vitamin or amino acid, respectively). If its energy drops to zero, the agent dies. If its energy reaches a specified threshold, the agent divides. During division, both daughter cells get half of the energy present in the individual before the division. The mobile agents in our model do not occupy space, meaning that in theory two agents can be at the same coordinates at the same time.

When a specified total amount of agents is reached, a transfer is simulated. This means a certain amount of agents is diluted into a fresh world. The amount of agents that triggers a transfer is calculated from the initial number of agents considering five population doublings. A transfer means that a number of agents is picked at random and randomly placed into an empty world (no vitamins, no amino acids). The rest of the agents are discarded (die). The transferred number of agents is equal to the total number of agents the model was initialized with at the start of the simulation. By default, the model does not have an additional source of vitamins and amino acids.

We do not intent to recreate the biology of the ALE experiment with the model but rather want to assess the possible fates of single mutants in such a scenario qualitatively. Therefore, the input values used for simulation are only in parts constrained by observations made in the experiments. We set the remaining inputs set to reasonable numbers. In the wet lab experiments, the species frequencies before a transfer were observed to be similar to the frequency in the inoculum, with about 8 times more bacteria than yeast. Since the relative amount stays roughly constant, we assume that the doubling times of yeast and bacteria are similar. We therefore set the energy threshold that triggers doubling in the model to equal values for bacterial and yeast agents and chose the conversion factors between metabolites and energy to result in a net gain of 1 unit energy per agent per time step under optimal growth conditions. Moreover, we conclude from the relation in the amounts that one yeast seems to be able to support around 8 bacteria and set the number of secreted metabolites per agent per time step accordingly. Appendix Table S8 contains all user defined model inputs and a short description, together with the values used to generate the simulations summarized in Fig EV4 (default values).

### Model simulation

To execute the model files, Netlogo needs to be downloaded and installed (https://ccl.northwestern.edu/netlogo/). The supplementary material includes two model files in the Netlogo format. One is a version of the model that implements all the features of the model and can be started and stopped by the user via a button (file Konstantinidis_ALE.nlogo). The second version of the model (file Konstantinidis_ALE_loop.nlogo) was used to obtain the simulated data summarized in Fig EV4. This version of the model contains a loop to simulate 200 replicates of a 25-transfer experiment for each value of the input *mixing* (0–1 in increments of 0.1). In this model version, the simulation cannot be stopped manually, but stops only after all runs have been completed. For this version of the model, it is recommended to untick the "view updates" option in the Netlogo menu "Interface tab" to speed up the simulation. After each run, the population dynamics are exported to a separate csv file (2,200 files in total per setup). To simulate the outcome for different values for the initial number of mutants and/or different fold changes of secretion, the user needs to change the respective inputs manually before the simulation is started. More information on how to use the models is available in the Netlogo menu "Info" tab of the model files.

To create Fig EV4, the csv files were imported into MATLAB and the relevant information was extracted and plotted in MATLAB (version 2018a). We did not check the results extracted from the simulation replicates for quantitative convergence. That means that additional simulations would probably lead to slightly different values for the assessed quantities (percentage of successful invasions, fraction of mutants within the total bacterial population). Nevertheless, we expect this to be quantitative differences only and not to change the observed patterns qualitatively, since the patterns were already visible after the first 100 simulations.

### Statistical analysis

Unless otherwise stated, *P*-values for comparisons between conditions were estimated using an unpaired two-tailed *t*-test. Statistical details of the experiments can be found in the figure legends.

# Data availability

The proteomic mass spectrometry data are deposited to the ProteomeXchange Consortium (http://proteomecentral.proteomexchange.org) via the PRIDE (Perez-Riverol *et al*, 2019) partner repository with the dataset identifier PXD022250 (https://www.ebi.ac.uk/pride/archive/projects/PXD022250).

The metabolomic data are available in MetaboLights (Haug *et al*, 2020, https://www.ebi.ac.uk/metabolights) with the database identifier MTBLS2217 (https://www.ebi.ac.uk/metabolights/MTBLS2217/) and in Mendeley (https://data.mendeley.com/drafts/53yyysbsf9).

The sequencing data are available in the European Nucleotide Archive (ENA, https://www.ebi.ac.uk/ena/) under accession number PRJEB39663 (https://www.ebi.ac.uk/ena/browser/view/PRJEB39663).

The model is available in BioModels with the database identifier MODEL2106100001 (https://www.ebi.ac.uk/biomodels/MODEL2106100001).

**Expanded View** for this article is available online.

## Acknowledgements

We acknowledge the support of the following core facilities at the European Molecular Biology Laboratory (Heidelberg, Germany): Metabolomics (R. Gathungu), Genomics (V. Benes and R. Hercog), Proteomics (R. Mattel and F. Stein), Flow Cytometry (M. Paulsen and D. Ordonez) and Advanced Light Microscopy (S. Reither). We thank S. Blasche for providing the parental bacterial isolates used in this study. This project has received funding from the European Research Council (ERC) under the European Union's Horizon 2020 research and innovation programme (Grant agreement no. 866028).

## Author contributions

DK and KRP conceived the project, designed experiments and wrote the manuscript. DK performed the experiments and data analysis. FP contributed to project planning, to sample preparation for the multi-omics experiments and to data analysis. E-MG performed the simulation study and wrote the corresponding manuscript text. KG contributed to the evolution and the cell surface experiments. EK contributed to GC-MS sample preparation, data acquisition and analysis. PJ performed the genome-scale metabolic model simulations. YK contributed to the selection of vitamin producing bacterial strains. SD and MZ contributed to the LC-MS analysis of the amino acids. KRP oversaw the project and helped with data interpretation.

## Conflict of interest

D.K., F.P., Y.K. and K.R.P. are co-inventors of a pending patent application (Patent No. 18188495.8-1120).

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
