## [Review Process File · Molecular Systems Biology]

Adaptive laboratory evolution of microbial co-cultures for improved metabolite secretion

Dimitrios Konstantinidis, Filipa Pereira, Eva-Maria Geissen, Kristina Grkovska, Eleni Kafkia, Paula Jouhten, Yongkyu Kim, Saravanan Devendran, Michael Zimmermann, and Kiran Patil

DOI: [10.15252/msb.202010189](https://doi.org/10.15252/msb.202010189)

Corresponding author: Kiran Patil (kp533@cam.ac.uk)

Review Timeline:

Submission Date:	22nd Dec 20
Editorial Decision:	4th Feb 21
Revision Received:	30th Apr 21
Editorial Decision:	4th Jun 21
Revision Received:	2nd Jul 21
Accepted:	7th Jul 21

Editor: Maria Polychronidou

Transaction Report:

4th Feb 2021

RE: MSB-2020-10189, Adaptive laboratory evolution of microbial co-cultures for improved metabolite secretion

Thank you again for submitting your work to Molecular Systems Biology. We have now heard back from the three referees who agreed to evaluate your study. Overall, the reviewers acknowledge that the study seems potentially interesting. They raise however a series of concerns, which we would ask you to address in a major revision.

I think that the recommendations of the reviewers are rather clear and I therefore do not see the need to repeat the points listed below. All issues raised by the reviewers need to be satisfactorily addressed. As you may already know, our editorial policy allows in principle a single round of major revision, so it is essential to provide responses to the reviewers' comments that are as complete as possible. Please contact me in case you would like to discuss in further detail any of the issues raised.

Reviewer #1:

Konstantinidis et. al. present an elegantly designed and well executed work in which they use adaptive lab evolution to increase the production of riboflavin by *Lactobacillus plantarum*. The authors used a cross-feeding community to couple increased production of riboflavin to the fitness on the producing species. This strategy was demonstrated by the authors to be successful for additional target molecules like folate, as well as for additional strains of lactic acid bacteria (*Lactococcus lactis* and *Brevibacterium casei*). The genetic, proteomic and metabolomic analyses provide compelling insights to the underlying evolved mechanism by which the target molecule production was increased. This work greatly contributes to the field of adaptive lab evolution by tackling one of the more challenging avenues of ALE - producing desired molecules. The system described in this paper will be instrumental for a wide range of follow-up work both basic and applicative. However, the manuscript can be improved by addressing the following comments:

1. A crucial parameter of the experimental design - conducting evolution in conditions that allow aggregates of bacteria and yeast is mentioned only in the discussion (Lines 369-379). Since in a well-mixed environment a mutant cell with increased (costly) production of cross-feeding molecules is not expected to increase in proportion, it would be better to inform the reader earlier in the text how the ALE was designed to overcome this issue. I found myself very distracted by this as I was reading the manuscript.

More generally, could the authors confirm that there was an increase in secretion per cell? (ie that the increase in the vitamins was not simply a result of the fact that the evolved LAB were better able to use the amino acids and hence grow more? The sequencing results seemed to indicate that the primary evolutionary pressure was on amino acid utilization...

2. Figure 1 could be clearer - in sub figure A it is clear that the computer cartoon of the lactic acid bacteria is to show that its genome has been verified to have the genes needed for riboflavin production. Yet, it is less clear what the computer cartoon of the yeast represents.

In sub figure B (legend) the evolved cultures are referred to as population while in the main text they are referred to as communities. It would be helpful for the reader to keep a clear distinction throughout the text between communities, populations and isolates. For example, the names of the 12 evolved communities are not specified and the relation to isolates derived from these evolved communities, like B4,C2,E6 and G7 (from Sub figures D&E) are not clear. Furthermore, these isolates are referred to as strains (subfigure D) while it seems they should be referred to as isolates or isolated colonies etc. (this is also applicable for the rest of the text and figures).

In sub figure C the authors should consider a simple distribution of the values from the 134 isolates as it might be more informative (provide more data) than their density plot (this comment might be applicable for additional figures that show density plots).

3. Identified mutations in the evolved bacterial isolates can be further analyzed. Are there any "intersections" between mutations that occurred in parallel evolved lines? Which similar genes/pathways/regulators/functions have mutated in parallel lines? To this end it might be useful to re-order the mutation table (Table S1) and cluster mutations in similar genes from different lines (if there were any)

4. Line 98: stain instead of strain

Reviewer #2:

Summary

Konstantinidis et al. co-evolved lactic acid bacteria (LAB) *Lactobacillus plantarum*, a species which naturally produces and secretes the vitamins riboflavin and folate, with yeast *Saccharomyces cerevisiae* that has been mutated to be auxotrophic for these vitamins, respectively. In addition, the yeast secretes amino acids for which the LAB are auxotrophic. After the evolution, the authors focus on the phenotypic and genetic changes in the LAB. In the riboflavin adaptive evolution, selected evolved strains secrete more riboflavin or accumulate higher amounts of flavin coenzymes. Whole genome sequencing in four evolved strains did not reveal a clear link to increased vitamin production, however proteomics analysis showed a higher abundance in Rib proteins involved in riboflavin biosynthesis, and exometabolome analysis confirmed an increased release of riboflavin, but no other changes compared to the parental strain. Intracellular levels of amino acids associated with riboflavin production were increased or are provided by the yeast. The adaptive evolution method was used to increase production of another vitamin, was specific to the mutualism model, and could be applied to other LAB species. This work provides a new model of adaptation that could potentially be applied to increase the production of other metabolites of interest.

General remarks

There are many instances where there is a lack of rigor (i.e. replicates) in the data shown and conclusions are drawn based on a single data point for a sample with no standard deviation/statistics shown. In addition, generalizations are made for multiple samples (strains) when the conclusion may only apply to a subset of the samples shown. In their conclusions, the authors frequently group all the evolved isolates together, despite the fact that they show differing phenotypes in several assays, which may be due to differences in the mutations present in these isolates. Parsing the phenotypic data alongside the WGS data might provide insights into which mutations underlie specific phenotypic changes seen in the isolates. The specific instances of lack of replicates and conclusions applied broadly are denoted below.

The adaptive evolution utilized in this study increases production of vitamin for a subset of strains, and the approach and findings would be advantageous for future engineering of microbes to increase production of goods. However, much of the data shown is difficult to interpret, the significant proteomics and metabolomics data should be represented in the main text or supplementary material, some of the claims are made broadly when it is not true for all the samples tested, and the WGS data is not parsed to identify the mutations that may underlie the phenotypes seen.

Major points

1. Line 82-86. Figure 1c. a) While many isolates have increased riboflavin secretion, it seems a proportion have decreased extracellular riboflavin compared to the parental control. What is the explanation or rationale for this result? Could this be because these isolates are "cheaters" from the evolution - that is these LAB are also benefiting from the increased secretion of riboflavin from other strains and themselves have reduced production of the vitamin, thus avoiding the metabolic costs? b) In addition, what is the variation in measurements for the parental strain? The data shown here is 'fold-change' compared to the parental strain, but it is not clear what the spread would be for multiple independent replicate measurements of the parental strain - similar numbers from a non-evolved population of the parental should be shown for comparison since the fold-change is shown. c) The methods suggest that the fluorescence was measured once for each isolate and then compared to the parental. For a robust comparison, riboflavin should be measured in triplicate (at least) for each strain, and compared to similar measurements for the parental strain, since there may be variation in measurements, as suggested by other panels in Figure 1. Without

these data, it is difficult to support the author's statement that 'nearly 60% of the isolates exhibited improved riboflavin secreting phenotype'. The same should be done for the data shown in Fig 3c and 3d.

2. Line 86-91. Figure 1d. Was there a specific reason why these four strains were selected for further analysis? This should be described in the text. Also, since there is some variability in the extracellular riboflavin secreted, standard deviations should be included in the averages listed in the text.

3. Line 101-106. Figure 1e. Earlier in the text (Line 94-95) it is stated that FAD was secreted at similar levels in the parental and evolved strains, but then later it is stated that C2 and G7 have secreted higher levels of FAD. Also, based on fold-change it is not clear they accumulated more riboflavin (amount increased?) although intracellular FAD looks higher. Here also the variability associated with these measurements is not shown, and it is not clear whether the levels were measured more than once for the evolved strains. These measurements should be performed in triplicate and the standard deviations should be shown. Minor point: it would also be helpful to view the fold-change=1 line to indicate no change from the parental strain across the figure.

4. Lines 136 - 138. Figure 2a. The authors state that both 'B4 and E6 exhibited a strong decrease in their growth rate in comparison to the parental', but in the figure, E6 appears to be similar to WT. Perhaps the authors could show a table with the values, or show a magnified inset with all the datapoints that are currently on top of each other.

5. Lines 138-140. Figure 2b. Similar to figure 2a, the authors state in general terms that the evolved bacterial strains performed better than the parental strain under low nitrogen conditions, but in the figure it looks like only B4 may be significantly different than the parental strain.

6. Line 141-142. Figure 2c. Co-cultures with three of the evolved strains grew better, but one did not (either B4 or D5, difficult to see due to the similar light blue colors of the lines). General remarks are given for all of the evolved strains, when one or more may have a different phenotype.

7. Line 143-144. Figure 2d. Strain D5 did not support 1.5-fold more yeast cells than the parental strain, and it's unclear whether the differences seen with B4 and C2 are significantly different from the parental.

8. Line 149-162. The authors look at cell-associated traits of the evolved strains and observe differences in auto- and co-aggregation as well as biofilm in monoculture. However, they do not demonstrate that "direct contact with yeast is not necessary for vitamin production". The decrease in aggregation could still provide enough contact to facilitate vitamin transfer. This claim is not supported until the next section (Line 163-179) with growth in the supernatant from parental or evolved strains (or yeast).

9. Line 153-154. Figure S1a. This shows aggregates with E6 and delrib4/5 yeast, but images of aggregation with parental LAB and delrib4/5 should be shown for comparison.

10. Line 157-159. Figure S1b. These panels are showing aggregation, with the y-axis label being "Aggregation Percentage (%)", and the data showing a decrease over time. But according to the methods and phenotype, there is an increase in aggregation over time rather than a loss which is what is shown here. The y-axis label should be changed to reflect that, such as the % of initial OD. Also, it is difficult to see significant differences between the evolved strains and the parental in these graphs, even though the authors state so.

11. Line 157-161. Figure S1c. In the text, the strains B4, C2, and E6 are mentioned and it is stated that these three strains formed more biofilm than in monoculture, however E6 is not shown in this figure for biofilm formation.

12. Line 167-169. Figure 2f. Here the evolved strains survive better in the yeast conditioned medium suggesting they adapted to grow in the presence of the secreted amino acids? However, how is this result reconciled with the increased growth of the evolved strains B4 and G7 in medium with limited supplementation of amino acids in Figure 2b?

13. Lines 169 - 174. The authors again state in general that increased fitness is displayed in the

conditioned media from the evolved yeast isolates, or from E6, but the isolates vary widely, and it would be useful to note which isolates showed a difference, and which did not.

14. Line 211-223. In the following lines, the way it is stated it is not clear if the transcription regulators and amino acid transporters are increased or decreased. Due to the interesting results with differentially abundant proteins that are transcriptional regulators and amino acid transporters (are they the same ones as in the WGS?) and proteins involved in the biosynthesis of purines and riboflavin, including RibA, the significantly different proteins should be represented as a figure/table in the main text, and/or a supplementary table with the fold-change in abundance for all strains, and the p-values obtained while testing for significant differences. How does the increased abundance of certain Rib proteins correlate with each respective strain's extracellular/intracellular levels of riboflavin, FAD, and FMN?

15. Line 225-246. The metabolomics data should be represented in a table/graph and referenced in the respective results section, similar to the proteomics data.

16. Line 232. Do any of the increased intracellular amino acids correlate with specific genes/proteins from the whole genome sequencing analysis/proteomics?

17. Lines 239 - 241. Figure S2c. The authors should show either the individual measurements or error bars showing the standard deviation of the replicates. Without any information about the variation in independent biological replicates, it is impossible to conclude whether the evolved strains are in fact over- or under-secreting specific amino acids.

18. Line 255. Add standard deviation for the average measurement of the best secreting isolate (B8) and also mention the parental average measurement of secreted folate.

19. Lines 258 - 260. Figure S4c. Only some of the evolved isolates show better growth, but the authors again generalize their conclusions to all isolates.

20. Lines 272 - 274. Fig 3c. The authors state that 'the maximum detected increase for any isolate from the monoculture evolution was only 1.2-fold, an order of magnitude lower than that observed in co-evolution', but in the figure the maximum for the monoculture evolution appears to have a \log_2 (fold-change) = ~ 1.5 , compared to the co-culture evolution which appears to have a maximum of \log_2 (fold-change) = $\sim 2.5-3$. So it is not clear what the authors are referring to.

21. Line 299-303. Figure S6 is referenced, but this is for Figure S7. The data would be better represented if the amount of vitamin in milk was subtracted to just show the contribution from the parental and evolved strains. Also, multiple replicates (with appropriate statistics, standard deviations) should be shown to demonstrate the variability and whether the evolved strains effectively add more vitamin. As it is presented, the differences shown appear very slight, and unlikely to be significant.

22. Line 362-363. A majority of strains is mentioned, but there's still approximately 60:40 ratio of overproducers to underproducers.

23. Line 382-385. Since there were so many mutations listed for each strain from the whole genome sequencing, it would be best to determine the specific mutations that are necessary/required for the increased vitamin production and introduce those to a wild isolate (such as the parental) rather than "readily" use strains that have many mutations with unknown consequences. Analyzing the phenotypic differences between the evolved strains in conjunction with the differences in observed mutations may aid in such a determination. However, the ALE method is robust in identifying the pathways and these strains may be safe for consumption.

24. Line 574. The citation for the CDM is a review. The original LTEE study (Lenski et al. 1991) which this review cites uses a glucose containing minimal medium. Which rich medium was used for the CDM design? And which previously designed media recipe components were combined?

25. Line 1237-1238. Table S1. The common genes with mutations among the strains should be denoted in the table (such as BGV74_00535 MarR in C2, E6, G7). Is the complete list of mutations with high frequency provided/available? Are any of the genes involved in amino acid uptake worth mentioning in the main text (Line 204) if they have been shown to uptake the specific amino acids

needed from the yeast to support LAB growth in the co-culture conditions? Do the high producers have any mutations/genes/gene-associations in common that may be the reason why they produce more vitamin?

Minor points

1. Figure 1a. Define $S_v=0$ and engineered yeast ($\Delta\text{rib4}/\text{rib5}$) and LAB riboflavin+ should be labeled in the illustration.
2. Figure 2a and b. Difficult to see the different symbols when overlapped. Some different color outlines or shading may help distinguish these samples.
3. Figure 2e. Legend shows 5 different samples/colors, but the figure has 7 different colored lines. Also, the error bars overlap making it difficult to see the variation for the individual lines - perhaps the authors could give the actual values for the slopes and maximal ODs to show the differences between the parental and the evolved strains.
4. In several figures (e.g. 2c, 2e, 2f, S4c), the line colors are extremely similar to each other, making it difficult to distinguish between them - these should be changed.
5. Line 281. Add "a" before "small increase"
6. Line 566. Define the abbreviations for these media (ex: De Man, Rogosa, and Sharpe (MRS))
7. Line 569. "triplicated" to "triplicate"

Reviewer #3:

In this manuscript entitled: "Adaptive laboratory evolution of microbial co-cultures for improved metabolite secretion", Konstantinidis et al., performed adaptive laboratory evolution (ALE) of pairwise cultures of lactic acid bacteria (LAB) with the yeast *Saccharomyces cerevisiae*. Species are mutualistic obligate and the growth of the co-culture relies on the cross-feeding of vitamins (riboflavin or folate) from the LAB to the auxotrophic yeast and the cross-feeding of amino acids from the yeast to the LAB. Here, they demonstrate that evolution of such community leads to the specific increased production of one cross-fed metabolite (here the vitamins), despite its fitness cost for the producing species (here the LAB), and to the improved growth of the overall community. First, Konstantinidis et al., focus on a co-culture between a Riboflavin producing strain of *Lactobacillus plantarum* and a *S. cerevisiae* strain auxotrophic to Riboflavin. They show that co-evolution leads to higher growth for the community and to the emergence of *L. plantarum* strains that produce higher concentrations of Riboflavin and that thus are able to support higher yeast growth. The authors also evolve *L. plantarum* in monoculture and showed that the pressure associated with cross-feeding was directly and specifically determining the evolution output associated with higher Riboflavin production. To reveal the underlying mechanisms, they perform a multi-omic analysis of *L. plantarum* evolved strains. While genome analysis didn't reveal direct mutations in riboflavin biosynthesis pathways, differential proteomic and metabolomic analysis between the evolved strains and the parental strains revealed modifications associated with the upregulation of the supply of riboflavin precursors in the evolved strains.

Using ALE experiments on different co-cultures of LAB and *S. cerevisiae*, the authors showed that their approach and conclusions are generalizable to other cross-fed metabolites (co-culture between folate producing *L. plantarum* strain and *S. cerevisiae* strain auxotrophic to folate) and to other LAB species (co-culture between riboflavin producing *Lactococcus lactis* or *Brevibacterium casei* and *S. cerevisiae* auxotrophic to riboflavin). Finally, they show that specific traits acquired through ALE in co-culture are conserved outside of the context of the evolution and potentially in application relevant set-ups.

Altogether, Konstantinidis et al., have produced a very interesting work with high-quality data. They

convincingly demonstrate the power of ALE in mutualistic co-cultures to generate novel strains with specific traits of interest that could not be obtained in ALE of monocultures. Applications of such technique are numerous and would span multiple biotechnological fields. Overall, the quality of the text and the figures are good and the flow of the story is easy to follow.

Below, I have included several comments and suggestions for improvements, that I hope will be constructive:

Major points

C1- In their work, the authors mention and use evolved yeast strains as well. While I understand that the focus is on the evolved *L. plantarum* strains capable of producing higher levels of vitamins, I think that adding a small paragraph dedicated to the specific description of the selection of evolved yeasts and their phenotype is required to improve the characterization of the ALE. Also, it will facilitate the navigation of the results when the authors mention these evolved yeasts strains in the condition media experiments for instance.

C2- Generally, when the authors show that a co-culture involving one evolved strain grows better than the community of parent strains, it is unclear whether this results from an improved growth of both species or only from the improved growth of only one of the community members.

Can the authors show the individual growth of each community member during a co-culture?

This would allow to see whether both members grow better in the community compared to their growth in the community composed by the parent strains, and this would also potentially improve our understanding of the dynamics between the two mutualistic species.

C3- From L 141 to L147, the authors conclude that the evolved *L. plantarum* strains (i) support a higher load of yeast cells and (ii) that they have evolved to use the resources provided by the yeast more efficiently. Yet, what they mean by that and how they reach to these conclusions is not obvious to me. Could the authors explain more specifically what in their experiments and their results led them to these conclusions?

-First, does "support a higher yeast population" means that the author measured more yeast cells/per *L. plantarum* evolved cells than yeast cells/parental *L. plantarum* cells (and could they show that information)? Or does that simply mean that the yeast population is overall higher when co-cultured with an evolved *L. plantarum* strain versus with the parental strain (assuming that the *L. plantarum* population of the evolved strain can also be higher than the parental population).

-Then, can the authors more clearly explain how the combination of the results presented from L 134 to 144 lead to the conclusion that the evolved *L. plantarum* strain use the resources more efficiently?

C4- The authors have also used *in silico* models to simulate the co-evolution outputs. I believe that the authors should integrate that supplementary information as a single results section.

Minor Points:

C1- In the Introduction, while the description and the utility of Adaptive Laboratory Evolution is very clearly presented, the specific goal/aspect or question that this work is addressing is not presented as clearly. I believe the authors could elaborate a little more about the specific objective of their work and how it will contribute to improving their field of research. I believe this would facilitate the understanding of the work while possibly expanding the impact of their research.

C2- Generally, the results section could benefit from the addition of some method details to

facilitate the comprehension of the described results.

Particularly, some of this information found in the method could be helpful to have in the results:

- how many transfers were performed, when are cells transferred and why
- why and how flow cytometry was performed

C3- How were the 4 *L. plantarum* evolved strains chosen and why?

C4- Were the differential analysis of proteins and metabolites concentration between the parent and the evolved *L. plantarum* strains performed between monocultures or between co-cultures with the parental yeast strain?

C5- L 98: "The evolved stain B4" should be "strain"

Response to the Editor's and the Reviewers' Comments

We are grateful to the reviewers and the editor for constructive feedback on our manuscript. Their suggestions were very helpful in preparing the revision, which, we believe, has improved the presentation and clarity of our findings.

Summary of main changes

1. All figures have been revised to show individual data points from replicates/isolates wherever relevant.
2. Additional data included on bacterial and yeast cell counts in communities using FACS.
3. The results and the discussion on cheater simulations are moved earlier in the Results section.
4. Additional data analysis: We carried out an integrative flux balance analysis using constraints from the proteomics measurements. The results show that the proteomic changes in the evolved cells are consistent with the increased riboflavin production. Key flux predictions are shown in Figure 4.
5. The manuscript has been edited to improve clarity. A version showing all changes is included with the submission.

Point-by-point response

Reviewers Comments:

Reviewer #1 (Remarks to the authors):

Konstantinidis et. al. present an elegantly designed and well executed work in which they use adaptive lab evolution to increase the production of riboflavin by *Lactobacillus plantarum*. The authors used a cross-feeding community to couple increased production of riboflavin to the fitness on the producing species. This strategy was demonstrated by the authors to be successful for additional target molecules like folate, as well as for additional strains of lactic acid bacteria (*Lactococcus lactis* and *Brevibacterium casei*). The genetic, proteomic and metabolomic analyses provide compelling insights to the underlying evolved mechanism by which the target molecule production was increased. This work greatly contributes to the field of adaptive lab evolution by tackling one of the more challenging avenues of ALE - producing desired molecules. The system described in this paper will be instrumental for a wide range of follow-up work both basic and applicative. However, the manuscript can be improved by addressing the following comments:

→ We thank the reviewer for their positive comments and constructive suggestions for improvement.

1a. A crucial parameter of the experimental design - conducting evolution in conditions that allow aggregates of bacteria and yeast is mentioned only in the discussion (Lines 369-379). Since in a well-mixed environment a mutant cell with increased (costly) production of cross-feeding molecules is not expected to increase in proportion, it would be better to inform the reader earlier in the text how the ALE was designed to overcome this issue. I found myself very distracted by this as I was reading the manuscript.

→ We agree that the use of non-shaking growth conditions likely contributed to the increased aggregation, and thereby checked the population of the cheaters. Our decision to use non-shaking conditions was also motivated by the observation that the lactic acid bacteria used in the study prefer microaerobic conditions (and thereby grow worse under shaking conditions). We have now discussed this point early in the Results section as below:

“We next performed a serial transfer adaptive laboratory evolution experiment with twelve populations derived from the same parental cultures. Non-shaking culturing conditions were used to facilitate microaerobic conditions preferred by the parental LAB strain. The non-shaking growth condition was also expected to check the emergence and/or dominance of cheater cells (i.e. cells that profit from the common goods but do not contribute in return) through facilitating spatial organisation, which generally favours co-operators over cheaters (Stump et al, 2018).”

1b. More generally, could the authors confirm that there was an increase in secretion per cell? (ie that the increase in the vitamins was not simply a result of the fact that the evolved LAB were better able to use the amino acids and hence grow more? The sequencing results seemed to indicate that the primary evolutionary pressure was on amino acid utilization...

→ Our data indeed shows increased production per cell. All LCMS measurements were performed using single isolates from the evolved population, and the data is normalized to OD₆₀₀ values. Further, the samples for the omics analysis were collected at comparable growth stages, and under high amino acids supplementation wherein the parental and the evolved strains grow comparably (Figure 2A).

To further substantiate this point, we have included additional data on bacterial and yeast cell counts in co-cultures (new panels A and B in Figure EV1; reproduced below). The bacterial cell counts increased, albeit marginally, for the co-cultures involving the evolved isolates. Yet, the ratio of the bacterial to yeast cells did not change significantly; this attests to the intended mutualistic nature of the yeast-LAB relation whereby the growth of each species is tightly linked to the other. In this mutualistic scenario, the increase in total biomass (i.e. yeast + bacterial cell counts), as well as increase in the growth rate as observed during evolution, necessarily requires increase in secretion per cell.

The text has been updated as below.

“Per co-culture OD, the number of bacteria somewhat increased, albeit with considerable variation across replicates (Figure EV1A). However, the ratio of the bacterial to yeast cells did not change significantly (p-value > 0.05), in line with the mutualistic nature of the yeast-lactic acid bacteria relation whereby the growth of both species is tightly linked (Figure EV1B).”

Figure EV1. A) Bacterial cell events in fluorescence-activated cell sorting (FACS) in yeast-LAB co-cultures. The co-cultures used are the same as in Figure 2C, and collected at 84 h after inoculation (n=3 biologically independent samples). The counts are normalised by the OD600 of the co-culture at the sample collection time. B) Yeast to bacteria cell ratio in co-cultures. Yeast cell numbers were estimated using RFP positive events (shown in Figure 2D) and bacterial cell numbers were estimated using FACS as in A. (n=3 biologically independent samples).

We also carried out additional computational analysis by combining proteomics data with flux balance analysis. This supported increased flux to the riboflavin pathway in the evolved isolates.

“To put the proteomic changes in the context of the overall metabolic network, we used these as constraints in flux balance analysis and thereby estimated metabolic fluxes in the parental strain and the evolved isolates (Methods). This proteomics-integrated flux balance analysis predicted increased flux to the riboflavin pathway for all the evolved isolates, showing that the observed changes in protein abundance are consistent with the increased vitamin production.”

And in the Discussion “Mechanistically, genomic, proteomic, flux balance and metabolomic analyses revealed concordant changes in the regulation of vitamin biosynthesis and precursor supply (Figure 4).”

Figure 4. Key metabolic changes in *L. plantarum* following coevolution with yeast. Bacterial pathways associated with the biosynthesis of riboflavin and its main precursors, and amino acids secreted by yeast are shown.

Metabolites and proteins with at least a two-fold change in abundance and FDR-adjusted p-value < 0.05 are highlighted.

2a. Figure 1 could be clearer - in sub figure A it is clear that the computer cartoon of the lactic acid bacteria is to show that its genome has been verified to have the genes needed for riboflavin production. Yet, it is less clear what the computer cartoon of the yeast represents.

→ We agree and have accordingly revised Figure 1.

2b. In sub figure B (legend) the evolved cultures are referred to as population while in the main text they are referred to as communities. It would be helpful for the reader to keep a clear distinction throughout the text between communities, populations and isolates. For example, the names of the 12 evolved communities are not specified and the relation to isolates derived from these evolved communities, like B4,C2,E6 and G7 (from Sub figures D&E) are not clear. Furthermore, these isolates are referred to as strains (subfigure D) while it seems they should be referred to as isolates or isolated colonies etc. (this is also applicable for the rest of the text and figures). In sub figure C the authors should consider a simple distribution of the values from the 134 isolates as it might be more informative (provide more data) than their density plot (this comment might be applicable for additional figures that show density plots).

→ We apologize for this inconsistency and have accordingly corrected the manuscript text and figures. A clarifying note explaining the naming of the community/isolate IDs has also been added to the Material and Methods.

"The communities were numbered (1-12) and each single colony isolate was identified by a letter (A - G). For example, strain A1 denotes isolate 'A' from community 1. Since one of the communities (community 9) collapsed during the adaptive laboratory evolution experiment, the twelfth line of the plate was filled with random isolates from all the other communities to increase the number of screened isolates."

Furthermore, these IDs were also clarified in the results section; e.g. "B4 (isolate B from community 4)".

Figure 1C is now changed to a beeswarm plot.

In Figure 3, we aim to highlight the differences across strains and conditions (mono- vs co-cultures), and hence believe that the density plots of the fold-changes are more informative. We now included individual ratios in the density plots to provide a direct visualization of the variations in the data.

3. Identified mutations in the evolved bacterial isolates can be further analyzed. Are there any "intersections" between mutations that occurred in parallel evolved lines? Which similar genes/pathways/regulators/functions have mutated in parallel lines? To this end it might be useful to re-order the mutation table (Table S1) and cluster mutations in similar genes from different lines (if there were any).

→ We did identify SNPs occurring on the same genes in isolates from different evolved lines. Appendix Table S2 has now been re-organized to mark these.

4. Line 98: stain instead of strain

→ Corrected.

Reviewer #2 (Remarks to the authors):

Konstantinidis et al. co-evolved lactic acid bacteria (LAB) *Lactobacillus plantarum*, a species which naturally produces and secretes the vitamins riboflavin and folate, with yeast *Saccharomyces cerevisiae* that has been mutated to be auxotrophic for these vitamins, respectively. In addition, the yeast secretes amino acids for which the LAB are auxotrophic. After the evolution, the authors focus on the phenotypic and genetic changes in the LAB. In the riboflavin adaptive evolution, selected evolved strains secrete more riboflavin or accumulate higher amounts of flavin coenzymes. Whole genome sequencing in four evolved strains did not reveal a clear link to increased vitamin production, however proteomics analysis showed a higher abundance in Rib proteins involved in riboflavin biosynthesis, and exometabolome analysis confirmed an increased release of riboflavin, but no other changes compared to the parental strain. Intracellular levels of amino acids associated with riboflavin production were increased or are provided by the yeast. The adaptive evolution method was used to increase production of another vitamin, was specific to the mutualism model, and could be applied to other LAB species. This work provides a new model of adaptation that could potentially be applied to increase the production of other metabolites of interest.

→ We are grateful for the referee's suggestions to improve our manuscript and thank them for drawing our attention to the potential unclarities/ambiguities.

General remarks

There are many instances where there is a lack of rigor (i.e. replicates) in the data shown and conclusions are drawn based on a single data point for a sample with no standard deviation/statistics shown.

→ This has been largely due to the choice of presenting average fold changes (which we thought would be more easy to interpret), and partly due to incomplete description. We apologise for the resulting unclarity and have addressed this throughout the revised manuscript, in particular by directly showing the individual data points in the figures. Responses to the specific points are provided below.

In addition, generalizations are made for multiple samples (strains) when the conclusion may only apply to a subset of the samples shown. In their conclusions, the authors frequently group all the evolved isolates together, despite the fact that they show differing phenotypes in several assays, which may be due to differences in the mutations present in these isolates. Parsing the phenotypic data alongside the WGS data might provide insights into which mutations underlie specific phenotypic changes seen in the isolates.

→ We agree that pairing the observed bacterial phenotype to genetic changes would be interesting. However, this is not the goal of our present study. The purpose of the genomic and the other omics analyses included in this study is to complement the phenotypic changes observed in the co-evolution experiments, and not to uncover genotype-phenotype relations in the lactic acid bacteria (LAB). The latter is a task that will require a separate investigation and is outside the scope of this work. In this context, we note that the "gold standard" for establishing genotype-phenotype relations is to reproduce the phenotype through engineering of specific mutations in the parental strains; this is not a straightforward task since LAB strains are not easy to engineer at such precision, and the methods are highly strain-dependent. Further, as multiple mutations can act in concert, a combinatorial approach would need to be taken. Nevertheless, we did want to gain insights into the metabolic reorganization in the evolved strains – hence the multi-omics follow-up. We agree that carrying out similar multi-omics analyses for a number of evolved strains (to make any meaningful associations, at least 10-20 strains would be needed) would be insightful for understanding the regulation of vitamin biosynthesis in LAB – but this is outside the scope of our present work.

The specific instances of lack of replicates and conclusions applied broadly are denoted below.

→ Responses to these are provided below point-by-point.

The adaptive evolution utilized in this study increases production of vitamin for a subset of strains, and the approach and findings would be advantageous for future engineering of microbes to increase production of goods. However, much of the data shown is difficult to interpret, the significant proteomics and metabolomics data should be represented in the main text or supplementary material, some of the claims are made broadly when it is not true for all the samples tested, and the WGS data is not parsed to identify the mutations that may underlie the phenotypes seen.

→ To provide a better overview of proteomics and metabolomics data, we have included an additional figure (Figure EV3). We have also revised the Appendix Table S2 to clarify which evolved isolates carry which mutations and the commonalities. Regarding associations between the mutations and the phenotypes, please see our response to the second General Remark above.

Major points

1. Line 82-86. Figure 1c.

1.a While many isolates have increased riboflavin secretion, it seems a proportion have decreased extracellular riboflavin compared to the parental control. What is the explanation or rationale for this result? Could this be because these isolates are "cheaters" from the evolution - that is these LAB are also benefiting from the increased secretion of riboflavin from other strains and themselves have reduced production of the vitamin, thus avoiding the metabolic costs?

→ Emergence of cheaters is indeed the likely explanation. A directly related question is why the co-cultures did not collapse due to cheater dominance (only 1 of the 48¹ co-cultures collapsed). To answer this, we included modelling showing that the non-shaking culturing conditions likely contributed to the promotion of the co-operators and consequent community stabilization through spatial segregation. This point was, however, not clear in the manuscript and we have now addressed this in two ways:

i) We describe the rationale for using non-shaking culturing conditions early in the Results section.

"We next performed a serial transfer adaptive laboratory evolution experiment with twelve populations derived from the same parental cultures. Non-shaking culturing conditions were used to facilitate microaerobic conditions preferred by the parental LAB strain. The non-shaking growth condition was also expected to check the emergence and/or dominance of cheater cells (i.e. cells that profit from the common goods but do not contribute in return) through facilitating spatial organisation, which generally favours co-operators over cheaters (Stump et al, 2018)."

ii) The simulation results are now described earlier in the Results section (previously it was in Discussion).

1.b In addition, what is the variation in measurements for the parental strain? The data shown here is 'fold-change' compared to the parental strain, but it is not clear what the spread would be for multiple independent replicate measurements of the parental strain - similar numbers from a non-evolved population of the parental should be shown for comparison since the fold-change is shown.

¹ 12 for riboflavin, 12 for folate production, 12 with *Lactococcus lactis* and 12 with *Brevibacterium casei*

→ We have now changed the plot to directly show the ODs rather than the fold changes (Figure 1C, reproduced below), including those for the parental strain.

Fig 1C. The riboflavin secretion was assessed in the culture supernatants by measuring the fluorescence intensity (440 nm excitation /520 nm emission) after 72 h (n = 134 (evolved isolates) and 5 (parental) biologically independent samples) and is shown as Relative Fluorescence Units (RFU).

1.c The methods suggest that the fluorescence was measured once for each isolate and then compared to the parental. For a robust comparison, riboflavin should be measured in triplicate (at least) for each strain, and compared to similar measurements for the parental strain, since there may be variation in measurements, as suggested by other panels in Figure 1. Without these data, it is difficult to support the author's statement that 'nearly 60% of the isolates exhibited improved riboflavin secreting phenotype'. The same should be done for the data shown in Fig 3c and 3d.

→ The objective of the initial screening was a rapid and high-throughput assessment to select strains with increased secretion. Using replicates at this early screening stage would have logistically complicated this task without any added benefit. We note that all subsequent measurements for the selected isolates have been carried out using biological replicates. Regarding the statement of 60% isolates exhibiting improved secretion, this is justified due to the following:

i) the variation in the parental measurements is very low (new Figure 1C).

ii) higher vitamin secretion by the 4 selected isolates with increased fluorescence readout (B4, C2, E6 and G7), and decreased secretion by D5, was confirmed by LCMS analysis. Further, the variability of riboflavin UPLC measurements correlate well with fluorescence measurements (Table S1), further validating the initial screen.

The variation observed amongst the evolved population (Figure 1C) thus reflects the biological variation, which is not surprising for co-evolution experiments.

2. Line 86-91. Figure 1d. Was there a specific reason why these four strains were selected for further analysis? This should be described in the text. Also, since there is some variability in the extracellular riboflavin secreted, standard deviations should be included in the averages listed in the text.

→ The strains were selected for further characterization based on the initial screen (fluorometry) followed by UPLC quantification (Appendix Table S1).

The text has been modified to “*Four evolved bacterial isolates with increased fluorescence values were selected for further characterization. The improved phenotype of these isolates was verified using Ultra Performance – Liquid Chromatography (UPLC, Appendix Table S1). Further, the amount of produced riboflavin from the selected isolates and the parental strain, cultured as*

monocultures in amino acid supplemented growth medium, was quantified using Liquid Chromatography - Mass spectrometry (LC-MS) analysis."

The standard deviation values have been added to the text.

3a. Line 101-106. Figure 1e. Earlier in the text (Line 94-95) it is stated that FAD was secreted at similar levels in the parental and evolved strains, but then later it is stated that C2 and G7 have secreted higher levels of FAD.

→ We thank the referee for bringing this unclarity to our attention. The text has been updated to:

"FAD was secreted at similar levels by the parental strain and isolates B4 and E6"; and "However, these isolates secreted higher amounts of FAD (1.7 and 1.9-fold increase for C2 and G7 respectively; p-value < 0.05)".

3b. Also, based on fold-change it is not clear they accumulated more riboflavin (amount increased?) although intracellular FAD looks higher. Here also the variability associated with these measurements is not shown, and it is not clear whether the levels were measured more than once for the evolved strains. These measurements should be performed in triplicate and the standard deviations should be shown. Minor point: it would also be helpful to view the fold-change=1 line to indicate no change from the parental strain across the figure.

→ The riboflavin measurements are added to the text.

"...did not show improved riboflavin secretion in comparison to the parental strain (C2: 60.9 ± 20.7 ng/mL, G7: 18.2 ± 3.2 ng/mL)."

"and accumulated higher intracellular amount of riboflavin (C2: 11 ± 6.7 ng/mL, G7: 15 ± 16.6 ng/mL, Figure 1E)".

All measurements and statistics are included in Appendix Table S1. Figure 1 is also updated to show the measurements directly instead of the fold changes.

4. Lines 136 - 138. Figure 2a. The authors state that both 'B4 and E6 exhibited a strong decrease in their growth rate in comparison to the parental', but in the figure, E6 appears to be similar to WT. Perhaps the authors could show a table with the values, or show a magnified inset with all the datapoints that are currently on top of each other.

→ Figures 2A and 2B have been updated to include magnified insets showing the data points that overlap. Moreover, the text has been updated to include the calculated p-values *"(B4: p-value < 0.05, E6: p-value < 0.05)"*.

5. Lines 138-140. Figure 2b. Similar to figure 2a, the authors state in general terms that the evolved bacterial strains performed better than the parental strain under low nitrogen conditions, but in the figure it looks like only B4 may be significantly different than the parental strain.

→ We have updated the text in question to better highlight the growth differences between the parental strain and the evolved isolates under low nitrogen conditions. For this experiment, the focus is on the difference between the 2 conditions. The parental strain's growth decreased with the decrease of available amino acids, while that of the evolved isolates remained similar or improved, supporting the better performance after evolution.

"When we mimicked the co-culture conditions by culturing the evolved bacterial isolates under limited nitrogen availability (Ponomarova et al, 2017), the parental strain's OD₆₀₀ value and

growth rate significantly decreased in comparison to the high nitrogen availability conditions (p -value < 0.05), while all evolved isolates had similar or improved growth rate (Figure 2B)."

6. Line 141-142. Figure 2c. Co-cultures with three of the evolved strains grew better, but one did not (either B4 or D5, difficult to see due to the similar light blue colors of the lines). General remarks are given for all of the evolved strains, when one or more may have a different phenotype.

→ We have now included the names of the isolates next to the corresponding lines in Figure panels 2C, 2E and 2F. The isolate D5 is a 'negative control' and hence the worse growth performance is expected. We have updated the text as below to clarify this.

"Additionally, co-cultures of the parental yeast and improved bacterial isolates grew better than the co-cultures involving either only the parental strains or the evolved 'negative control' bacterial isolate D5 (isolate with decreased riboflavin production, Appendix Table S1, Figure 2C)."

7. Line 143-144. Figure 2d. Strain D5 did not support 1.5-fold more yeast cells than the parental strain, and it's unclear whether the differences seen with B4 and C2 are significantly different from the parental.

→ The strain D5 is a 'negative control' (which is now clarified in the manuscript text). For the other strains, we have included the statistical testing.

"Sorting of the co-culture populations showed that the evolved isolates B4, C2 and E6 supported 1.5 to 3.5-fold more yeast cells than their parental strain (p -value < 0.05, Figure 2D), while the isolate D5 (isolate with decreased riboflavin production) could still support a small number of yeast cells similar to the parental (Figure 2D)."

8. Line 149-162. The authors look at cell-associated traits of the evolved strains and observe differences in auto- and co-aggregation as well as biofilm in monoculture. However, they do not demonstrate that "direct contact with yeast is not necessary for vitamin production".

The decrease in aggregation could still provide enough contact to facilitate vitamin transfer. This claim is not supported until the next section (Line 163-179) with growth in the supernatant from parental or evolved strains (or yeast).

→ We agree and have modified the text to *"These experiments indicate that there was no selection pressure towards increased aggregation."*

9. Line 153-154. Figure S1a. This shows aggregates with E6 and delrib4/5 yeast, but images of aggregation with parental LAB and delrib4/5 should be shown for comparison.

→ Supplementary Figure 1 has been updated to Figure EV1. Panel C shows aggregates of the two parental strains (parental *L. plantarum* and parental Δ rib4:rib5), and panel D aggregates of bacterial isolate E6 and parental Δ rib4:rib5.

10a. Line 157-159. Figure S1b. These panels are showing aggregation, with the y-axis label being "Aggregation Percentage (%)", and the data showing a decrease over time. But according to the methods and phenotype, there is an increase in aggregation over time rather than a loss which is what is shown here. The y-axis label should be changed to reflect that, such as the % of initial OD.

→ Figure S1b has been updated to Figure EV1E, where the label of the Y-axis has been renamed as 'Decrease in OD₆₀₀'.

10b. Also, it is difficult to see significant differences between the evolved strains and the parental in these graphs, even though the authors state so.

→ We thank the referee for drawing our attention to this unclarity. Our mention of significant differences concerned the experiment of biofilm formation. To clarify this, the text has been updated to:

“Evolved isolates C2 and E6 could aggregate more than the parental strain, both in the presence of yeast or in monoculture, while isolate B4 was aggregating less, independently of the presence of yeast. Nevertheless, the observed differences were small and not statistically significant (Figure EV1E). Furthermore, all three isolates showed significantly higher tendency for biofilm formation in monoculture in comparison to their parental strain (p -value < 0.05), but this change was not observed in the corresponding co-cultures (Figure EV1F).”

11. Line 157-161. Figure S1c. In the text, the strains B4, C2, and E6 are mentioned and it is stated that these three strains formed more biofilm than in monoculture, however E6 is not shown in this figure for biofilm formation.

→ Typographic mistake of writing strain E6 as F6 has been corrected in Figure EV1F.

12. Line 167-169. Figure 2f. Here the evolved strains survive better in the yeast conditioned medium suggesting they adapted to grow in the presence of the secreted amino acids? However, how is this result reconciled with the increased growth of the evolved strains B4 and G7 in medium with limited supplementation of amino acids in Figure 2b?

→ The limited supplementation conditions resemble closer to the secreted amino acids levels (Ponomarova *et al.*, Cell Systems, 2017), hence the correspondence between the two scenarios. The text has been updated to clarify this:

*“When we mimicked the co-culture conditions by culturing the evolved bacterial isolates under limited nitrogen availability (Ponomarova *et al.*, 2017), the parental strain’s OD_{600} value and growth rate significantly decreased in comparison to the high nitrogen availability conditions (p -value < 0.05), while all evolved isolates had similar or improved growth rate (Figure 2B)”*

13. Lines 169 - 174. The authors again state in general that increased fitness is displayed in the conditioned media from the evolved yeast isolates, or from E6, but the isolates vary widely, and it would be useful to note which isolates showed a difference, and which did not.

→ As shown in Appendix Figure S1, all evolved yeast isolates clearly differentiate from the parental auxotrophic yeast strain. Similarly, as depicted in Figure EV2, both the parental *L. plantarum* and the evolved isolate E6 grow better when cultured in the conditioned medium of the evolved *S. cerevisiae* isolates. Since the main focus of our work was the improvement of vitamin secretion by *L. plantarum*, we restricted the description to the observed trend.

14 a. Line 211-223. In the following lines, the way it is stated it is not clear if the transcription regulators and amino acid transporters are increased or decreased

→ The data shows a mixed response, which also varies between isolates (now summarized in Figure EV3). For example, one of the transcription regulators (UNIPROT: F9USY4, Transcription regulator, Lacl family, maltose-related) is upregulated in isolates B4 and C2 and downregulated in isolates E6 and G7. Amongst the two oligo peptide transporters shown in Figure 4, one is upregulated (UNIPROT: F9UN55, Oligopeptide ABC transporter, ATP-binding protein) and the other downregulated (UNIPROT: F9USW2, Oligopeptide ABC transporter, substrate binding protein).

14b. Due to the interesting results with differentially abundant proteins that are transcriptional regulators and amino acid transporters (are they the same ones as in the WGS?) and proteins involved in the biosynthesis of purines and riboflavin, including RibA, the significantly different proteins should be represented as a figure/table in the main text, and/or a supplementary table with the fold-change in abundance for all strains, and the p-values obtained while testing for significant differences. How does the increased abundance of certain Rib proteins correlate with each respective strain's extracellular/intracellular levels of riboflavin, FAD, and FMN?

→ The current best *Lactobacillus plantarum* genome annotation is for strain WCFS1, which is a riboflavin auxotroph. For this reason, we had to resort to other genomes for riboflavin biosynthesis and transporters. Further, the annotations for transporters and transcription regulators are often unspecific (mostly the family they belong to, e.g. ABC transporter or oligo peptide transporter). In general, there is no overlap between genes with SNPs and change in the abundance of the corresponding proteins. This is not completely surprising as coding regions SNPs usually do not lead to expression changes. The only observed correspondence between SNP and differential protein expression is ABC transporter permease (gene BGV74_06015 and protein F9UNJ7, Appendix Table S2).

To better relate the proteomic changes with the observed phenotype, we now included a new computational analysis by combining proteomics data with metabolic modelling. This supported increased flux to the riboflavin pathway in all the tested evolved isolates and thus the text has been updated accordingly.

“To put the proteomic changes in the context of the overall metabolic network, we used these as constraints in flux balance analysis and thereby estimated metabolic fluxes in the parental strain and the evolved isolates (Methods). This proteomics-integrated flux balance analysis predicted increased flux to the riboflavin pathway for all the evolved isolates, showing that the observed changes in protein abundance are consistent with the increased vitamin production.”

In the Discussion:

“Mechanistically, genomic, proteomic, flux balance and metabolomic analyses revealed concordant changes in the regulation of vitamin biosynthesis and precursor supply (Figure 4).”

15. Line 225-246. The metabolomics data should be represented in a table/graph and referenced in the respective results section, similar to the proteomics data.

→ Intracellular metabolomics and proteomics data are summarized in Figure EV3. Moreover, all metabolomics data are available in the database Metabolights (<https://www.ebi.ac.uk/metabolights/>) with the dataset identifier MTBLS2217, and proteomics data have been deposited to the ProteomeXchange Consortium (<http://proteomecentral.proteomexchange.org>) via the PRIDE partner repository with the dataset identifier PXD022250.

16. Line 232. Do any of the increased intracellular amino acids correlate with specific genes/proteins from the whole genome sequencing analysis/proteomics?

→ Given the large number of proteins and metabolites in comparison with the isolates analyzed, it is not possible to infer any statistically meaningful associations. It is possible that the increased levels of glutamine may be linked with the increased abundance of glutamine transporter (Figure 4). However, the increased transporter abundance might also be associated with the observed change in glutamine secretion by the evolved *S. cerevisiae* isolates.

17. Lines 239 - 241. Figure S2c. The authors should show either the individual measurements or error bars showing the standard deviation of the replicates. Without any information about the variation in independent biological replicates, it is impossible to

conclude whether the evolved strains are in fact over- or under-secreting specific amino acids.

→ We have updated Figure S2c (now Figure EV2) to overlay the data points on to the bar plots.

18. Line 255. Add standard deviation for the average measurement of the best secreting isolate (B8) and also mention the parental average measurement of secreted folate.

→ The standard deviation and the parental average have been included in the main text.

“...the best secreting evolved isolate (B8) produced 190 ± 35.6 ng/mL of folate, while the parental strain secreted 48.3 ± 10.5 ng/mL of folate (Figure 3A).”

19. Lines 258 - 260. Figure S4c. Only some of the evolved isolates show better growth, but the authors again generalize their conclusions to all isolates.

→ We have updated the text as below:

“Similar to the riboflavin case, the parental folate auxotroph yeast ($\Delta abz1$) showed improved growth in the conditioned medium of the evolved bacterial isolates with improved secretion (B8 and H1), while all the evolved bacterial isolates grew better than their parental when cultured in the yeast conditioned medium (Appendix Figure S3B and C).”

20. Lines 272 - 274. Fig 3c. The authors state that 'the maximum detected increase for any isolate from the monoculture evolution was only 1.2-fold, an order of magnitude lower than that observed in co-evolution', but in the figure the maximum for the monoculture evolution appears to have a \log_2 (fold-change) = ~ 1.5 , compared to the co-culture evolution which appears to have a maximum of \log_2 (fold-change) = $\sim 2.5-3$. So it is not clear what the authors are referring to.

→ We thank the referee for drawing our attention to this unclarity, which arose from the smoothing / 'trailing tail' introduced by the density function. We have now included the data points on the abscissa of the density plots (Figure 3C and D). Only a single isolate exhibited \log_2 fold change of 1 (two-fold change). On the other hand, the maximum \log_2 fold change in co-culture was 2.45 (\sim five-fold change). The average \log_2 fold change for the monocultures and the co-cultures are 0.37 and 1.42. Top and bottom five values are shown in the table below for comparison.

\log_2 fold change in the fluorescence intensity (indicating riboflavin levels) of *L. plantarum* isolates compared to their parental strain.

	Co-culture evolved isolates	Monoculture evolved isolates	Isolates evolved for Folate production
	-2.5	-2.3	-2.3
	-2.1	-2.3	-2.3
	-1.9	-2.2	-2.2
	-1.86	-2.2	-2.2
	-1.8	-2.1	-2.1
	2.1	1.008	1
	1.9	0.9	0.9
	1.89	0.8	0.6
	1.85	0.7	0.5
	1.81	0.7	0.3

The text in question has been modified to:

“Moreover, the average increase for the isolates evolved in monoculture was only 1.2-fold, an order of magnitude lower than the changes observed in co-evolution (Figure 3C).”

21a. Line 299-303. Figure S6 is referenced, but this is for Figure S7.

→ Thank you, corrected.

21b. The data would be better represented if the amount of vitamin in milk was subtracted to just show the contribution from the parental and evolved strains. Also, multiple replicates (with appropriate statistics, standard deviations) should be shown to demonstrate the variability and whether the evolved strains effectively add more vitamin. As it is presented, the differences shown appear very slight, and unlikely to be significant.

→ Figure S7 (now Figure EV5) has been modified to show individual measurements and statistical comparisons.

22. Line 362-363. A majority of strains is mentioned, but there's still approximately 60:40 ratio of overproducers to underproducers.

→ >50% is generally regarded as a majority.

23. Line 382-385. Since there were so many mutations listed for each strain from the whole genome sequencing, it would be best to determine the specific mutations that are necessary/required for the increased vitamin production and introduce those to a wild isolate (such as the parental) rather than "readily" use strains that have many mutations with unknown consequences. Analyzing the phenotypic differences between the evolved strains in conjunction with the differences in observed mutations may aid in such a determination. However, the ALE method is robust in identifying the pathways and these strains may be safe for consumption.

→ We appreciate the reviewer's view that our ALE method is a robust way to identify/improve specific pathways and simultaneously create safe to consume strains.

As we discussed above in response to the general remarks, even though we agree that pairing the observed bacterial phenotype to specific genetic changes that occurred during evolution would be interesting, this is out of the scope of our present study.

24. Line 574. The citation for the CDM is a review. The original LTEE study (Lenski et al. 1991) which this review cites uses a glucose containing minimal medium. Which rich medium was used for the CDM design? And which previously designed media recipe components were combined?

→ The reference in line 438 about the CDM recipe is “Ponomarova, O. *et al.* Yeast Creates a Niche for Symbiotic Lactic Acid Bacteria through Nitrogen Overflow. *Cell Syst.* 5(4), 345-357 doi: 10.1016/j.cels.2017.09.002 (2017)”. The misunderstanding was perhaps due to the separate “Supplementary Information references”.

25a. Line 1237-1238. Table S1. The common genes with mutations among the strains should be denoted in the table (such as BGV74_00535 MarR in C2, E6, G7).

→ Appendix Table S2 has been updated to group SNPs occurring in the same genes across different isolates.

25b. Is the complete list of mutations with high frequency provided/available? Are any of the genes involved in amino acid uptake worth mentioning in the main text (Line 204) if

they have been shown to uptake the specific amino acids needed from the yeast to support LAB growth in the co-culture conditions? Do the high producers have any mutations/genes/gene-associations in common that may be the reason why they produce more vitamin?

→ None of the detected mutations are directly associated with the uptake of the yeast secreted amino acids, one of the reasons prompting us to undertake proteomics and metabolomics analyses. Indeed, these two omics provided a much clearer picture of the underlying network changes as we discuss in the manuscript (Figure 4 and the related text).

Minor points:

1. Figure 1a. Define Sv=0 and engineered yeast (del rib4/rib5) and LAB riboflavin+ should be labeled in the illustration.

→ Figure 1a has been updated accordingly and has been simplified to increase clarity.

2. Figure 2a and b. Difficult to see the different symbols when overlapped. Some different color outlines or shading may help distinguish these samples.

→ We have now included insets focusing on the areas where the symbols overlap. Furthermore, we have included the names of the isolates next to their respective growth lines in panels c, e and f of Figure 2.

3. Figure 2e. Legend shows 5 different samples/colors, but the figure has 7 different colored lines. Also, the error bars overlap making it difficult to see the variation for the individual lines - perhaps the authors could give the actual values for the slopes and maximal ODs to show the differences between the parental and the evolved strains.

→ Panel "e" of Figure 2 has been updated to improve the clarity and the visibility of the data.

4. In several figures (e.g. 2c, 2e, 2f, S4c), the line colors are extremely similar to each other, making it difficult to distinguish between them - these should be changed.

→ The color scheme is uniform throughout the different figures; to better differentiate the strains, we have included the name of each strain next to the corresponding line.

5. Line 281. Add "a" before "small increase"

→ Corrected.

6. Line 566. Define the abbreviations for these media (ex: De Man, Rogosa, and Sharpe (MRS))

→ Abbreviations have been defined for all the used media.

7. Line 569. "triplicated" to "triplicate"

→ Corrected.

Reviewer #3 (Remarks to the authors):

In this manuscript entitled: "Adaptive laboratory evolution of microbial co-cultures for improved metabolite secretion", Konstantinidis et al., performed adaptive laboratory evolution (ALE) of pairwise cultures of lactic acid bacteria (LAB) with the yeast *Saccharomyces cerevisiae*. Species are mutualistic obligate and the growth of the co-culture relies on the cross-feeding of vitamins (riboflavin or folate) from the LAB to the auxotrophic yeast and the cross-feeding of amino acids

from the yeast to the LAB. Here, they demonstrate that evolution of such community leads to the specific increased production of one cross-fed metabolite (here the vitamins), despite its fitness cost for the producing species (here the LAB), and to the improved growth of the overall community. First, Konstantinidis et al., focus on a co-culture between a Riboflavin producing strain of *Lactobacillus plantarum* and a *S. cerevisiae* strain auxotrophic to Riboflavin. They show that co-evolution leads to higher growth for the community and to the emergence of *L. plantarum* strains that produce higher concentrations of Riboflavin and that thus are able to support higher yeast growth. The authors also evolve *L. plantarum* in monoculture and showed that the pressure associated with cross-feeding was directly and specifically determining the evolution output associated with higher Riboflavin production. To reveal the underlying mechanisms, they perform a multi-omic analysis of *L. plantarum* evolved strains. While genome analysis didn't reveal direct mutations in riboflavin biosynthesis pathways, differential proteomic and metabolomic analysis between the evolved strains and the parental strains revealed modifications associated with the upregulation of the supply of riboflavin precursors in the evolved strains. Using ALE experiments on different co-cultures of LAB and *S. cerevisiae*, the authors showed that their approach and conclusions are generalizable to other cross-fed metabolites (co-culture between folate producing *L. plantarum* strain and *S. cerevisiae* strain auxotrophic to folate) and to other LAB species (co-culture between riboflavin producing *Lactococcus lactis* or *Brevibacterium casei* and *S. cerevisiae* auxotrophic to riboflavin). Finally, they show that specific traits acquired through ALE in co-culture are conserved outside of the context of the evolution and potentially in application relevant set-ups. Altogether, Konstantinidis et al., have produced a very interesting work with high-quality data. They convincingly demonstrate the power of ALE in mutualistic co-cultures to generate novel strains with specific traits of interest that could not be obtained in ALE of monocultures. Applications of such technique are numerous and would span multiple biotechnological fields. Overall, the quality of the text and the figures are good and the flow of the story is easy to follow. Below, I have included several comments and suggestions for improvements, that I hope will be constructive

→ We are thankful to the reviewer for their encouraging feedback and valuable suggestions for improving the manuscript.

Major points

1. In their work, the authors mention and use evolved yeast strains as well. While I understand that the focus is on the evolved *L. plantarum* strains capable of producing higher levels of vitamins, I think that adding a small paragraph dedicated to the specific description of the selection of evolved yeasts and their phenotype is required to improve the characterization of the ALE. Also, it will facilitate the navigation of the results when the authors mention these evolved yeasts strains in the condition media experiments for instance.

→ To clarify that we also isolated yeast cells from the evolved communities after the completion of the evolution experiment, we updated the text as follows.

*“From the evolved populations, we isolated both evolved *L. plantarum* (134 isolates) and *S. cerevisiae* (6 isolates) cells for further characterisation.”*

The evolved yeasts were used in the conditioned media assays and to evaluate their amino acid secreting phenotype. The text has been updated to:

“Furthermore, both the parental strain and the evolved bacterial isolates displayed improved fitness in the conditioned medium of the evolved yeast isolates; with 16 to 49% increase in the optical density (Figure EV2A and B). When these evolved yeast isolates were cultured in conditioned medium from the evolved bacterial isolate E6 (Appendix Figure S2B), they showed similar maximal optical density as the parental $\Delta rib4:rib5$ strain, but their growth rate increased on average by 34%.”

“Indeed, all the evolved yeast isolates secreted higher amounts of glutamate and alanine, but lower amounts of glutamine and serine (Figure EV2C).”

2. Generally, when the authors show that a co-culture involving one evolved strain grows better than the community of parent strains, it is unclear whether this results from an improved growth of both species or only from the improved growth of only one of the community members. Can the authors show the individual growth of each community member during a co-culture? This would allow to see whether both members grow better in the community compared to their growth in the community composed by the parent strains, and this would also potentially improve our understanding of the dynamics between the two mutualistic species.

→ While we did not follow growth kinetics of the individual species, our data supports improved growth of both species after evolution. The results from the FACS experiment (Figures 2D, EV1A and B) showing that the ratio between the two species remains similar, together with the observed increase in the community growth rate, indicate this. Moreover, both evolved bacteria and yeast isolates performed better than their respective parental strains during the conditioned media experiments (Figures 2E and F, EV2A and B and Appendix Figure S1A and B).

3a. From L 141 to L147, the authors conclude that the evolved *L. plantarum* strains (i) support a higher load of yeast cells and (ii) that they have evolved to use the resources provided by the yeast more efficiently. Yet, what they mean by that and how they reach to these conclusions is not obvious to me. Could the authors explain more specifically what in their experiments and their results led them to these conclusions?

-First, does "support a higher yeast population" means that the author measured more yeast cells/per *L. plantarum* evolved cells than yeast cells/parental *L. plantarum* cells (and could they show that information)? Or does that simply mean that the yeast population is overall higher when co-cultured with an evolved *L. plantarum* strain versus with the parental strain (assuming that the *L. plantarum* population of the evolved strain can also be higher than the parental population).

→ As seen in the FACS data (Figure EV1B), the ratio between yeast and bacteria cells varies, but this variability is not statistically significant (p -value > 0.05). Thus, our previous text was indeed misleading and we have updated it to:

“Per co-culture OD, the number of bacteria somewhat increased, albeit with considerable variation across replicates (Figure EV1A). However, the ratio of the bacterial to yeast cells did not change significantly (p -value > 0.05), in line with the mutualistic nature of the yeast-lactic acid bacteria relation whereby the growth of both species is tightly linked (Figure EV1B). By secreting higher amount of riboflavin, the evolved isolates provided more resources to their yeast partner, which, in turn, provided amino acids to the bacteria.”

3b. Then, can the authors more clearly explain how the combination of the results presented from L 134 to 144 lead to the conclusion that the evolved *L. plantarum* strain use the resources more efficiently?

→ The increased growth of co-culture between evolved lactic acid bacteria and the parental yeast (Figure 2C), and the improved growth of evolved LAB under limited nitrogen availability (Figure 2B), show the improved ability of LAB to assimilate the amino acids secreted by yeast. This conclusion is further supported by the conditioned medium assay, where all bacteria receive the same amount of yeast secreted amino acids and the evolved *L. plantarum* isolates show improved fitness compared to their parental strains (Figure 2F).

4. The authors have also used in silico models to simulate the co-evolution outputs. I believe that the authors should integrate that supplementary information as a single results section.

→ Following the referee's suggestion, the results section of the manuscript has been updated to include the key observations from the *in silico* simulations.

"We were able to capture the key aspects of the coevolution experiments in a simulation study using an agent-based model. In particular, we were interested in understanding why the cheaters, i.e. bacterial cells that secreted low amounts or no vitamin but utilized the common goods (amino acids secreted by yeasts), did not lead to the population collapse. We simulated the fate of the mutants with an improved secreting phenotype but slower growth and studied the effect of increased mixing of the liquid culture on the mutant's chance to persist. First, we simulated a scenario of genetic drift where the mutant secretes the same amount of vitamin as the wild type, to establish a baseline to which the other simulations can be compared. Next, we repeated this simulation setup for different degrees of increase in secretion. We observed that the chance of a better secreting mutant to invade, i.e. to still be present after 25 transfers, was significantly higher (Fisher's exact test, p -value < 0.05) for low mixing strengths. This was observed for all simulated degrees of increased secretion (Figure EV4A). In the absence of mixing, the chance of successful invasion was tenfold higher than for a mutant with increased secretion than expected by chance (Fisher's exact test, p -value < 0.05) (Figure EV4A). Nevertheless, with increasing mixing strength, the chance of invasion first dropped below the null expectation (Fisher's exact test, p -value > 0.05) and then to zero. The absolute values for the chance of invasion did not vary significantly between the mutants with different degrees of secretion increase (Figure EV4A). In addition to the chance of invasion, we assessed the fraction of the mutant in the total bacterial population at the end of the simulation in successful invasions. We found that this fraction depended on how strong the secretion was increased, with higher secretion resulting in lower fractions of the mutant. The mutant fraction also depended on the strength of mixing, with stronger mixing resulting in lower mutant fractions down to total extinction of the mutant bacteria. The simulation results corroborate that a mutant with increased secretion but slower growth can have a selection advantage in an obligate mutualistic community in the absence of mixing (Figure EV4B-E)."

Minor points:

1. In the Introduction, while the description and the utility of Adaptive Laboratory Evolution is very clearly presented, the specific goal/aspect or question that this work is addressing is not presented as clearly. I believe the authors could elaborate a little more about the specific objective of their work and how it will contribute to improving their field of research. I believe this would facilitate the understanding of the work while possibly expanding the impact of their research.

→ We are thankful towards the referee for bringing this unclarity to our attention. To address this, we updated the text as following:

"While this minimal requirement underlines the elegance and the success of adaptive laboratory evolution, it also underscores its limited applicability to the traits that impose a toll on the cell fitness, such as metabolite secretion."

"To enable improvement of fitness-costly metabolite secretion while keeping the advantages offered by adaptive laboratory evolution, we here used mutualistic cross-feeding to exert selection pressure for increased production of the target compound."

"The lactic acid bacteria strains, which are not engineered, as well as the target products, riboflavin and folate, are relevant for food biotechnological applications. Beyond this direct industrial relevance, our study establishes a proof of concept for the feasibility of improving fitness-costly traits using mutualistic communities."

- 2a. Generally, the results section could benefit from the addition of some method details to facilitate the comprehension of the described results. Particularly, some of this information found in the method could be helpful to have in the results:

- how many transfers were performed, when are cells transferred and why.

→ We agree and have updated the text as below.

“We next performed a serial transfer adaptive laboratory evolution experiment with twelve populations derived from the same parental cultures. Non-shaking culturing conditions were used to facilitate microaerobic conditions preferred by the parental LAB strain. The non-shaking growth condition was also expected to check the emergence and/or dominance of cheater cells (i.e. cells that profit from the common goods but do not contribute in return) through facilitating spatial organisation, which generally favours co-operators over cheaters (Stump et al, 2018). In the evolution experiment, cultures were transferred to fresh media whenever a community reached the stationary stage of growth. Following 25 transfers, the maximal optical density had increased on average sevenfold (Figure 1B).”

2b. Why and how flow cytometry was performed.

→ We have updated the text to:

“Sorting of the co-culture populations showed that the evolved isolates B4, C2 and E6 supported 1.5 to 3.5-fold more yeast cells than their parental strain (p -value < 0.05, Figure 2D), while the isolate D5 (isolate with decreased riboflavin production) could still support a number of yeast cells similar to the parental (Figure 2D).”

3. How were the 4 *L. plantarum* evolved strains chosen and why?

→ Selection of the *L. plantarum* isolates was based on the initial fluorescence screen and HPLC measurements of riboflavin, FMN and FAD. As per the referee’s suggestion the text has been updated to:

“Four evolved bacterial isolates with increased fluorescence values were selected for further characterization. The improved phenotype of these isolates was verified using Ultra Performance Liquid Chromatography (UPLC, Appendix Table S1). Further, the amount of produced riboflavin from the selected isolates and the parental strain, cultured as monocultures in amino acid supplemented growth medium, was quantified using Liquid Chromatography - Mass spectrometry (LC-MS) analysis.

4. Were the differential analysis of proteins and metabolites concentration between the parent and the evolved *L. plantarum* strains performed between monocultures or between co-cultures with the parental yeast strain?

→ Both proteomics and metabolomics analysis were performed in bacterial monocultures supplemented with amino acids. The text is updated to:

“Through proteomics analysis of bacterial monocultures under amino acid supplementation,” and “Gas Chromatography - Mass Spectrometry (GC-MS) analysis of the exometabolome, originating from the same bacterial monocultures that used for the proteomics analysis”.

5. L 98: "The evolved stain B4" should be "strain"

→ Corrected.

RE: MSB-2020-10189R, Adaptive laboratory evolution of microbial co-cultures for improved metabolite secretion

Thank you for sending us your revised manuscript. We have now heard back from the three reviewers who were asked to evaluate your study. Overall, the reviewers think that the study has improved as a result of the performed revisions. However, as you will see below, reviewers #2 and #3 still raise some remaining concerns, which we would ask you to address in a last round of revision. It is especially important to address the remaining points of reviewer #2 and provide some further clarifications regarding their concerns on the data analysis and interpretation.

We would also ask you to address some editorial issues listed below.

Reviewer #1:

The authors have done an excellent and detailed job of responding to the questions and concerns raised by me and the other referees. I believe that the manuscript is now an excellent addition to MSB, and should be published as is.

Reviewer #2:

This manuscript describes the experimental co-evolution of obligate mutualistic relationships between bacteria and yeast, leading to increased production of metabolically costly vitamins. This is a resubmission of a previous manuscript, and the authors have addressed several of the previous comments and concerns, leading to a much-improved manuscript with datasets, and individual data points being explicitly shown now, and some nuance in describing the variation in phenotypes of the different evolved isolates.

However, there still remain a few instances where the authors generalize their results for all isolates, even though the data only shows the phenotype for a few. Further, some of the authors' claims are not supported by their data, and so should be changed. My specific concerns are:

1. Appendix table 1 now shows the data for the measurement of both intracellular and extracellular riboflavin from the parental strain as well as several evolved isolates, which is useful to evaluate the changes in riboflavin production upon experimental evolution, the central idea of the manuscript. However, the data raise several questions.

- Why are the measurements so different between UPLC and LCMS ? In the UPLC values, there is only a very slight difference in extracellular riboflavin production between the parental strain and the evolved isolates, which would negate the very premise of the study.

- The fluorescence data is very different from both the UPLC and HPLC data, in that the fluorescence shows significant increases for all 4 evolved isolates - B4, C2, E6, G7. Given that C2 and G7 do not show increased extracellular riboflavin by either of the other 2 methods, how accurate is the fluorescence assay ?

- Minor point : The legend to Appendix 1 says there are six evolved isolates instead of five. Also, the authors can mention that D5 is a negative control, earlier in the text, when they refer to Appendix Table 1.

2. Page 5, lines 23-26: The text states that C2 and G7 accumulated higher intracellular amount of riboflavin. However, the measurements : 11 ± 6.7 , and 15 ± 16.6 do not appear significantly different from the parental (10.3 ± 1.7) due to the high variability in the isolate measurements.

3. Page 7, lines 9-11: "Furthermore, both the parental strain and the evolved bacterial isolates displayed improved fitness in the conditioned medium of the evolved yeast isolates; with 16 to 49% increase in the optical density (Figure EV2A and B)." The figure legend states that A refers to growth of the parental strain and B to a single isolate E6, not multiple evolved bacterial isolates. The authors should explicitly say so.

4. Page 7, lines 11-14: The figure referred to should be S1, instead of S2B. Further, it is stated that the evolved yeast isolate showed similar maximal optical density as the parental strain, but in figure S1, the maximal OD of the evolved isolates appears to be significantly higher than the parental.

5. Page 8, lines 3-12 : Figure EV3. It is not clear what the annotations of the proteins are. Which proteins listed are the transcriptional regulators and amino acid transporters mentioned in the text? A table with the protein designations, annotations, and abundance values may serve as a better representation of the data.

6. Further, in figure EV3, the color scheme is not very clear, so several claims that the authors make in the text don't appear to be reflected in the figure.

- Page 8 lines 27-29: Ribose-5-phosphate and Glucose-6-phosphate don't look to be increased in the figure.

- Page 8 lines 30-31: Levels of glutamate (glutamic acid) don't appear to be increased in the figure.

As mentioned in point 5 above, these data will be better represented in a table with the gene names, annotations, and abundance values.

7. Page 8, line 37 - page 9, line 2: "Indeed, all the evolved yeast isolates secreted higher amounts of glutamate and alanine, but lower amounts of glutamine and serine (Figure EV2C)." The statistics results for Figure EV2C-H are not described. Are all of the evolved yeast strains secreting

significantly more alanine, and less serine? The levels of serine, in particular, do not appear to be significantly different between the parental strain and several of the isolates. Further, it appears the isolates are also secreting higher aspartate, but this is not mentioned.

8. Page 9, lines 15-17 : "When grown in monocultures, in medium supplemented with amino acids, the evolved isolates grew to higher maximum OD600 than their parental strain, and also exhibited higher growth rates (Figure 3B)." Isolates B8 and H1 do not appear to show increased growth rate compared to the parental strain. It should be specified which strains show the phenotype, instead of generalizing.

9. In Figure EV1F : What are all the isolates shown ? They are never mentioned in the text, and having so many isolates in the figure makes it difficult to visualize the isolates that are discussed in the text. Similarly, in Figure S2B and C, the extra isolates shown are never discussed in the text.

10. Page 9, lines 35-36 : Based on the authors' explanation (average log₂ fold-change for monocultures is 0.37 and for co-cultures is 1.42), it appears that the difference is 2-fold, which may be what the authors meant by 'an order of magnitude'. However, the authors should explicitly say 2-fold, because an order of magnitude can be misconstrued to mean 10-fold.

11. Page 11, lines 16-19. It is stated that the evolved isolates increased riboflavin concentration in milk, but so did the parental. The authors mention the p-values < 0.05 making it appear that B4 produces significantly more riboflavin compared to the parental, but it is actually significantly more than milk, and not statistically different from the parental. Even isolate E6 shows only a very slight difference compared to the parental. It is better to mention the results comparing the evolved strains vs. the parental strain here, and also discuss this caveat in the discussion where the authors mention that their evolved isolates retain their secreting phenotype in milk (page 13 lines 4-8).

12. The authors mention the Harcombe et al PNAS 2018 paper in the context of evolution of metabolic exchange, but do not explicitly state that that manuscript demonstrates the experimental co-evolution of secretion of costly metabolites. Perhaps the authors could discuss in the introduction how their study adds to this previous one.

Minor comments :

1. Page 9, lines 11-12 : Figure should be Appendix Fig. S2A; similar on page 9 line 21 it should be Fig. S2B and C.

2. Page 9 line 32 : Figure should be Appendix Fig. S3A.

3. Page 10 line 14 : Figure should be Appendix figure S3B and C.

Reviewer #3:

In their revised version of the manuscript "Adaptive laboratory evolution of microbial co-cultures for improved metabolite secretion" Konstandinidis et al., have addressed all my comments, and I believe, the other reviewers' comments as well. This new version shows increased quality and higher clarity for the presented results. Thus, I would like to thank the authors for their hard work and congratulate them for this improved version and very interesting work.

After reading this new version, I still have 3 minor comments that should be easy to address:

1- The authors should add the statistical tests they used to generate p-values presented in this new version (it is only mentioned in the Community simulation section).

2- The authors have added Flux Balance Analysis in this new manuscript. As this elegantly complements their results, I believe the authors should add some Methods information directly in the Results section when Flux Balance Analysis is mentioned. More specifically, they should:

- Mention that they use MARGE in the Results section (it will be easier to connect that section to the associated Methods part)

- Describe in 1 sentence what Flux Balance Analysis and what MARGE generally do (and mention their objective function here as well)

3- The title of the current section "Co-culture simulation" could be improved to be more informative. This could be something like: "Agent-based model simulations of co-cultures support the fitness advantage of increased secretion in obligate mutualistic community"

Response to the Editor's and the Reviewers' Comments

We are thankful to the reviewers and the editor for the constructive feedback on our manuscript.

Point-by-point response

Reviewers Comments:

Reviewer #1 (Remarks to the authors):

The authors have done an excellent and detailed job of responding to the questions and concerns raised by me and the other referees. I believe that the manuscript is now an excellent addition to MSB, and should be published as is.

→ We are thankful to the reviewer for their positive feedback.

Reviewer #2 (Remarks to the authors):

This manuscript describes the experimental co-evolution of obligate mutualistic relationships between bacteria and yeast, leading to increased production of metabolically costly vitamins. This is a resubmission of a previous manuscript, and the authors have addressed several of the previous comments and concerns, leading to a much-improved manuscript with datasets, and individual data points being explicitly shown now, and some nuance in describing the variation in phenotypes of the different evolved isolates.

However, there still remain a few instances where the authors generalize their results for all isolates, even though the data only shows the phenotype for a few. Further, some of the authors' claims are not supported by their data, and so should be changed. My specific concerns are:

→ We apologise for the unclarity, and impreciseness concerning the isolate phenotype descriptions and thank the reviewer for bringing these to our notice. Please see below our point-by-point response.

Major points

1. Appendix table 1 now shows the data for the measurement of both intracellular and extracellular riboflavin from the parental strain as well as several evolved isolates, which is useful to evaluate the changes in riboflavin production upon experimental evolution, the central idea of the manuscript. However, the data raise several questions.

1a. Why are the measurements so different between UPLC and LCMS ? In the UPLC values, there is only a very slight difference in extracellular riboflavin production between the parental strain and the evolved isolates, which would negate the very premise of the study.

→ Reported UPLC measurements are Area Under Curve (AUC, arbitrary units), and thus the numbers should not be directly compared with the LCMS data, which is based on calibration against standards and thus reported in concentration units. The reason for using different techniques is that they have different advantages. While UPLC is good for quantitation, LCMS is better for the molecular identity (and often more sensitive). Use of different methods showing the same trend is a plus (in fact, multiplication from a statistical confidence point of view). We note that all the comparisons are performed between the measurements from the same technique.

1b. The fluorescence data is very different from both the UPLC and HPLC data, in that the fluorescence shows significant increases for all 4 evolved isolates - B4, C2, E6, G7. Given that C2 and G7 do not show increased extracellular riboflavin by either of the other 2 methods, how accurate is the fluorescence assay ?

→ The use of fluorescence assay was motivated by its easy applicability to screening. Selected isolates were further characterised by UPLC and LC-MS assays as detailed in the manuscript. Like any screening method, the fluorescence assay has its pluses and minuses, hence we made the effort to perform the measurements with additional techniques. Nevertheless, the fluorescence method does well in distinguishing different production levels, as can be seen in the figure below.

Comparison between UPLC and fluorescence readouts. The scatterplot represents the average value of biological triplicates for the UPLC analysis and a single measurement with the fluorescence assay.

We further note that the multiple measurements performed on biological replicates of the parental strain were consistent ('Parental column' in Figure 1C, copied below).

1C. The riboflavin secretion was assessed in the culture supernatants by measuring the fluorescence intensity (440 nm excitation /520 nm emission) after 72 h (n = 134 (evolved isolates) and 5 (parental) biologically independent samples) and is shown as Relative Fluorescence Units (RFU).

Finally, we would like to highlight that our observations are followed-up with in-depth molecular analyses (genomics, proteomics) that support the overall conclusions.

1c. Minor point : The legend to Appendix 1 says there are six evolved isolates instead of five. Also, the authors can mention that D5 is a negative control, earlier in the text, when they refer to Appendix Table 1.

→ Thank you for the suggestion. The legend of Appendix 1 has been updated.

The text in the manuscript has been updated to mention isolate D5 earlier in the text:

“Four evolved bacterial isolates with increased fluorescence values, as well as one isolate with non-improved phenotype (D5) were selected for further characterization. The phenotype of these isolates was verified using Ultra Performance Liquid Chromatography (UPLC, Appendix Table S1). Further, the amount of produced riboflavin from the isolates with improved phenotype and the parental strain.”

2. Page 5, lines 23-26: The text states that C2 and G7 accumulated higher intracellular amount of riboflavin. However, the measurements : 11 +/- 6.7, and 15 +/- 16.6 do not appear significantly different from the parental (10.3 +/- 1.7) due to the high variability in the isolate measurements.

→ The corresponding text has been updated to *“However, these isolates secreted higher amounts of FAD (1.7 and 1.9-fold increase for C2 and G7 respectively; p-value < 0.05), and had similar levels of intracellular riboflavin as the parental strain (C2: 11 ± 6.7 ng/mL, G7: 15 ± 16.6 ng/mL, Figure 1E).”*

3. Page 7, lines 9-11: "Furthermore, both the parental strain and the evolved bacterial isolates displayed improved fitness in the conditioned medium of the evolved yeast isolates; with 16 to 49% increase in the optical density (Figure EV2A and B)." The figure legend states that A refers to growth of the parental strain and B to a single isolate E6, not multiple evolved bacterial isolates. The authors should explicitly say so.

→ The corresponding text has been updated to *“Furthermore, both the parental strain and the evolved bacterial isolate E6 displayed improved fitness in the conditioned medium of the evolved yeast isolates, with 16% and 49% increase in the optical density, respectively (Figure EV2A and B).”*

4. Page 7, lines 11-14: The figure referred to should be S1, instead of S2B. Further, it is stated that the evolved yeast isolate showed similar maximal optical density as the parental strain, but in figure S1, the maximal OD of the evolved isolates appears to be significantly higher than the parental.

→ The corresponding text has been updated to *“When these evolved yeast isolates were cultured in conditioned medium from the parental *L. plantarum* (Appendix Figure S1A) and the evolved bacterial isolate E6 (Appendix Figure S1B), they exhibited increased maximal optical density compared to the parental *Δrib4:rib5* strain, as well as increased growth rate (~34% on average).”*

5. Page 8, lines 3-12 : Figure EV3. It is not clear what the annotations of the proteins are. Which proteins listed are the transcriptional regulators and amino acid transporters mentioned in the text? A table with the protein designations, annotations, and abundance values may serve as a better representation of the data.

→ To address this, Appendix Table S3 has been added listing the proteins used in Figure EV3, including Uniprot identifiers, database annotations, and the observed difference in abundance in comparison to the parental strain.

6. Further, in figure EV3, the color scheme is not very clear, so several claims that the authors make in the text don't appear to be reflected in the figure.

→ Tables S3 and S4 have been added in the Appendix for increased clarity.

6a. Page 8 lines 27-29: Ribose-5-phosphate and Glucose-6-phosphate don't look to be increased in the figure.

→ Table S4 has been added in the Appendix listing the metabolites and the fold changes. In more detail, the increase for Ribose-5-phosphate ranged from 0.76 log₂fold change (1.7 fold-change) for isolate B4, to 1.16 log₂fold change (2.2 fold-change) for isolate G7. The observed increase was the same for isolates C2 and E6, circa 0.8 log₂fold change (1.75 fold-change).

Glucose-6-phosphate's increase ranged from 1.02 log₂fold change (isolate E6, 2.03 fold-change) to 1.52 log₂fold change (isolate C2, 2.87 fold-change). The observed log₂fold change for isolate B4 was 1.05 (2.07 fold-change) and 1.5 (2.82 fold-change) for G7.

6b. Page 8 lines 30-31: Levels of glutamate (glutamic acid) don't appear to be increased in the figure.

→ The levels of glutamate (Table S4) increased by 0.97 log₂fold change (1.96 fold-change) for isolate B4, 1.84 log₂fold change for isolate C2, 1.7 log₂fold change (3.25 fold-change) for isolate E6 and 1.78 log₂fold change (3.44 fold-change) for isolate G7.

6c. As mentioned in point 5 above, these data will be better represented in a table with the gene names, annotations, and abundance values.

→ Table S4 has been added in the Appendix for increased clarity.

7. Page 8, line 37 - page 9, line 2: "Indeed, all the evolved yeast isolates secreted higher amounts of glutamate and alanine, but lower amounts of glutamine and serine (Figure EV2C)." The statistics results for Figure EV2C-H are not described. Are all of the evolved yeast strains secreting significantly more alanine, and less serine? The levels of serine, in particular, do not appear to be significantly different between the parental strain and several of the isolates. Further, it appears the isolates are also secreting higher aspartate, but this is not mentioned.

→ The corresponding text has been updated to *"Indeed, all the evolved yeast isolates secreted higher amounts of glutamate and aspartate (two-tailed t-test, p-value < 0.05, Figure EV2D and G), with two of the isolates (EY1 and EY5) secreting higher amounts of alanine as well (two-tailed t-test, p-value ≤ 0.05, Figure EV2C). On the other hand, all isolates secreted lower amounts of glutamine (two-tailed t-test, p-value < 0.05, Figure EV2F), while the levels of threonine and serine remained largely unchanged (Figure EV2E and H)."*

Moreover, Figure EV2C-H has been updated to include the *p-value* for each isolate, in comparison to the parental *Saccharomyces cerevisiae* strain.

Amino acid secretion. Amino acid measurements (LCMS analysis) of extracellular samples from the parental $\Delta rib4:rib5$ *S. cerevisiae* and 6 evolved yeast isolates. Shown are the areas under curve (AUC) from the corresponding peaks. Bar heights mark average; dots show individual data points; n=3 biological replicates. P-values based on unpaired two-tailed t-test.

8. Page 9, lines 15-17 : "When grown in monocultures, in medium supplemented with amino acids, the evolved isolates grew to higher maximum OD600 than their parental strain, and also exhibited higher growth rates (Figure 3B)." Isolates B8 and H1 do not appear to show increased growth rate compared to the parental strain. It should be specified which strains show the phenotype, instead of generalizing.

→ The text has been updated to "When grown in monocultures, in medium supplemented with amino acids, all the evolved isolates grew to higher maximum OD₆₀₀ than their parental strain, and isolates A3 and C9 also exhibited higher growth rates (Figure 3B)."

9a. In Figure EV1F : What are all the isolates shown ? They are never mentioned in the text, and having so many isolates in the figure makes it difficult to visualize the isolates that are discussed in the text.

→ Figure EV1F is updated to show individual data points and only isolates described in the main manuscript have been included.

Biofilm formation. Estimation of the light absorbance at 570 nm from dye bound in the biofilm of bacterial monoculture (left), or bacteria co-culture with the parental auxotrophic yeast (right) Bar heights mark average; dots show individual data points; $n=4$ biological replicates. P -values for comparisons between conditions were estimated using an unpaired two-tailed t-test.

9b. Similarly, in Figure S2B and C, the extra isolates shown are never discussed in the text.

→ Figure S2 has been updated to exclude the isolates that are not discussed in the main text.

Appendix Figure S2. Co-evolution for the increased production of folate. A) Twelve populations (*L. plantarum* + $\Delta abz1$ *S. cerevisiae* strain), deriving from the same parental cultures, were evolved in parallel in 96well plates. The growth of each community improved similarly for every replicate, based on Optical Density measurements. In the figure are the dark black line represents the mean OD_{600} values of the 12 populations. **B)** Growth kinetics of the parental and evolved bacteria strains in conditioned medium from wild-type *S. cerevisiae* ($n=3$ biologically independent samples; data are presented as means \pm s.d – grey bars). **C)** Growth kinetics of the parental $\Delta abz1$ *S. cerevisiae* strain in conditioned medium of the parental and evolved bacteria strains ($n=3$ biologically independent samples; data are presented as means \pm s.d – grey bars).

10. Page 9, lines 35-36 : Based on the authors' explanation (average log₂ fold-change for mono-cultures is 0.37 and for co-cultures is 1.42), it appears that the difference is 2-fold, which may be what the authors meant by 'an order of magnitude'. However, the authors should explicitly say 2-fold, because an order of magnitude can be misconstrued to mean 10-fold.

→ We apologise for the unclarity. The difference between monoculture and co-culture is indeed 2-fold. The corresponding text has been updated to *“Moreover, the average increase for the isolates evolved in monoculture was only 1.2-fold, twice lower than the change observed in coevolution (Figure 3C).”*

11. Page 11, lines 16-19. It is stated that the evolved isolates increased riboflavin concentration in milk, but so did the parental. The authors mention the p-values < 0.05 making it appear that B4 produces significantly more riboflavin compared to the parental, but it is actually significantly more than milk, and not statistically different from the parental. Even isolate E6 shows only a very slight difference compared to the parental. It is better to mention the results comparing the evolved strains vs. the parental strain here, and also discuss this caveat in the discussion where the authors mention that their evolved isolates retain their secreting phenotype in milk (page 13 lines 4-8).

→ Both B4 (p = 0.15) and E6 (p<0.05) show increase in comparison to the parental. The p-values are now mentioned in the main text.

“While riboflavin concentration increased by 1.25-fold in cultures of the parental strain, the evolved isolates B4 and E6 showed 1.35 and 1.46-fold increase (B4: two-tailed t-test, p-value < 0.05, compared to milk control and p-value = 0.15 compared to the parental strain, E6: two-tailed t-test, p-value < 0.05 compared to milk control, p-value < 0.05 compared to the parental), respectively (Figure EV5).”

The trend exhibited by the two strains is supportive of our conclusion: while the p-value for B4 is “only” 0.15, the two observations together statistically strengthen the conclusion (probability that both observations would be by chance is 0.0075). We would also like to note the B4 and E6 are distinct from each other and outcomes of an inherently variable process (adaptive evolution). Thus, it is neither necessary nor expected that all evolved strains display identical phenotypes, and therefore both of tested strains exhibiting the increase in milk is notable.

12. The authors mention the Harcombe et al PNAS 2018 paper in the context of evolution of metabolic exchange, but do not explicitly state that that manuscript demonstrates the experimental co-evolution of secretion of costly metabolites. Perhaps the authors could discuss in the introduction how their study adds to this previous one.

→ The difference of our study is now discussed early in the introduction

“These studies primarily focus on establishing microbial models of cross-feeding, with less emphasis on the molecular basis of adaptations. Further, while amino acid cross-feeding is common in natural communities (Machado et al., 2021) and has also been the basis of synthetic communities (Wintermute and Silver, 2010), cross-feeding involving other nutrients has been less well studied.”

Also, now in discussion we highlight how the combined findings of ours and the previous studies support the generality of increased secretion of costly metabolites as a favored evolutionary strategy.

“Additionally, we show that the selection pressure for increased production is linked to the presence of the mutualistic partner and not with the absence of the compound from the

environment. Our results, together with previous observations of amino acid exchange (Mee et al., 2014), demonstrate that increased secretion of costly metabolites is an exploitable strategy for adaptive laboratory evolution."

Minor comments :

1. Page 9, lines 11-12 : Figure should be Appendix Fig. S2A; similar on page 9 line 21 it should be Fig. S2B and C.

→ Corrected

2. Page 9 line 32 : Figure should be Appendix Fig. S3A.

→ Corrected

3. Page 10 line 14 : Figure should be Appendix figure S3B and C.

→ Corrected

Reviewer #3:

In their revised version of the manuscript "Adaptive laboratory evolution of microbial co-cultures for improved metabolite secretion" Konstandinidis et al., have addressed all my comments, and I believe, the other reviewers' comments as well. This new version shows increased quality and higher clarity for the presented results. Thus, I would like to thank the authors for their hard work and congratulate them for this improved version and very interesting work.

→ We are thankful to the reviewer for their encouraging feedback and suggestions to further improve the manuscript.

After reading this new version, I still have 3 minor comments that should be easy to address:

1. The authors should add the statistical tests they used to generate p-values presented in this new version (it is only mentioned in the Community simulation section).

→ All the statistical tests have been added where appropriate.

2. The authors have added Flux Balance Analysis in this new manuscript. As this elegantly complements their results, I believe the authors should add some Methods information directly in the Results section when Flux Balance Analysis is mentioned. More specifically, they should:

2a. Mention that they use MARGE in the Results section (it will be easier to connect that section to the associated Methods part).

2b. Describe in 1 sentence what Flux Balance Analysis and what MARGE generally do (and mention their objective function here as well).

→ The Results section has been updated as follows:

"To put the proteomic changes in the context of the overall metabolic network, we used the genome-scale metabolic model of L. plantarum for estimating the metabolic fluxes in the parental strain and the evolved isolates that best fit to the relative metabolic enzyme levels (Methods). Specifically, we performed a Metabolic Analysis with Relative Gene Expression

(MARGE, Methods) which utilizes linear optimization to predict optimal metabolic states that best fit the relative changes in protein abundances, while the total metabolic proteome level is maintained. Through MARGE we predicted increased flux to the riboflavin pathway for all the evolved isolates, showing that the observed changes in the metabolic enzyme abundances are consistent with the increased vitamin production."

3. The title of the current section "Co-culture simulation" could be improved to be more informative. This could be something like: "Agent-based model simulations of co-cultures support the fitness advantage of increased secretion in obligate mutualistic community".

→ We agree. The section's title has been updated to:

'Agent-based model supports fitness advantage of increased secretion'.

Manuscript number: MSB-2020-10189RR, Adaptive laboratory evolution of microbial co-cultures for improved metabolite secretion

Thank you again for sending us your revised manuscript. We think that the performed revisions have satisfactorily addressed the remaining issues of the reviewers. As such, I am pleased to inform you that your paper has been accepted for publication

Corresponding Author Name: Kiran Raosaheb Patil

Manuscript Number: MSB-2020-10189